# Mapping the molecular and structural specialization of the skin basement membrane for inter-tissue interactions

Ko Tsutsui [1], Hiroki Machida [1,2,3], Asako Nakagawa[1], Kyungmin Ahn[1], Ritsuko Morita [1], Kiyotoshi Sekiguchi[4], Jeffrey H. Miner [5] & Hironobu Fujiwara [1,2,3✉]

Inter-tissue interaction is fundamental to multicellularity. Although the basement membrane (BM) is located at tissue interfaces, its mode of action in inter-tissue interactions remains poorly understood, mainly because the molecular and structural details of the BM at distinct inter-tissue interfaces remain unclear. By combining quantitative transcriptomics and immunohistochemistry, we systematically identify the cellular origin, molecular identity and tissue distribution of extracellular matrix molecules in mouse hair follicles, and reveal that BM composition and architecture are exquisitely specialized for distinct inter-tissue interactions, including epithelial–fibroblast, epithelial–muscle and epithelial–nerve interactions. The epithelial–fibroblast interface, namely, hair germ–dermal papilla interface, makes asymmetrically organized side-specific heterogeneity in the BM, defined by the newly characterized interface, hook and mesh BMs. One component of these BMs, laminin α5, is required for hair cycle regulation and hair germ–dermal papilla anchoring. Our study highlights the significance of BM heterogeneity in distinct inter-tissue interactions.

[1] Laboratory for Tissue Microenvironment, RIKEN Center for Biosystems Dynamics Research (BDR), Kobe, Japan. [2] Graduate School of Science and Technology, Kwansei Gakuin University, Sanda, Japan. [3] Graduate School of Medicine, Osaka University, Suita, Japan. [4] Laboratory of Matrixome Research and Application, Institute for Protein Research, Osaka University, Suita, Japan. [5] Division of Nephrology, Department of Medicine, Washington University School of Medicine, St. Louis, MO, USA. ✉email: hironobu.fujiwara@riken.jp

The extracellular matrix (ECM) is a complex noncellular network of multicellular organisms that plays essential roles in animal development and homoeostasis. The basement membrane (BM) is a thin sheet-like ECM located at tissue borders, where it compartmentalizes and also tightly integrates tissues[1,2]. The BM has several crucial roles: (1) providing structural support to cells that is essential for the development of organ structure; (2) signalling to cells through adhesion receptors including integrins; (3) controlling the tissue distributions and activities of soluble growth factors; and (4) endowing tissue mechanical characteristics[3–5]. Thus, the composition and structure of the BM are critical in many vital phenomena, including developmental patterning, inter-tissue interactions and stem cell niche formation.

The BM is composed of numerous molecules exhibiting spatiotemporal expression patterns during development and homoeostasis, indicating that individual cell types are exposed to tailor-made BM niches[6–8]. In mammals, the entire set of ECM molecules, called the matrixome or matrisome, is encoded by ~300 ECM genes[7,9] (http://matrisomeproject.mit.edu/). Although information about the unique distribution, biochemical activities and in vivo functions of individual BM molecules has been accumulated, the entire molecular landscape of the BM, including its cellular origins, tissue localizations and pattern-forming processes, in any organ remains largely unknown. One major reason for this lies in the biochemical properties of ECM proteins, including their large size, insolubility and crosslinked nature. This has impeded systematic comprehensive characterization of ECM specialization at the cellular resolution.

Mouse hair follicle (HF) is an excellent model to investigate the formation and function of spatiotemporally specialized ECMs because this mini-organ is tiny, yet has clear epithelial and dermal compartments associated with specific tissue architecture and functions (Fig. 1a)[10,11]. Another prominent feature of the HF is its regenerative ability. HFs undergo cycles of regeneration (anagen), regression (catagen) and resting (telogen) phases during hair regeneration to continuously supply new hairs[12]. This regenerative process is underpinned by epithelial stem cells residing in the basal layer around the HF bulge region. Different epithelial stem cell types are compartmentalized along the longitudinal axis of the HF[10,11]. These different cells are associated with distinct dermal cells, such as the lanceolate mechanosensory complexes in the upper bulge (UB) for tactile sensation and epithelial stem cell regulation[13,14], arrector pili muscle in the mid-bulge (MB) for piloerection[15] and dermal papilla (DP) in the hair germ (HG) for HF development and regeneration[16]. At anagen phase onset, the DP activates primed stem cells in the HG, leading to regeneration of the HF's lower cyclic portion. All of these epithelial–dermal interactions occur via the BM.

Previous studies suggested that the BM is an important niche component for both epithelial stem cells and dermal cells. Loss of contact with the ECM or reduced integrin expression triggers the differentiation of cultured epidermal stem cells[17]. Deletion of the transmembrane protein collagen XVII (COL17), cytoplasmic integrin-linked kinase or kindlin, which mediate the linkage between epidermis and BM, resulted in defective epidermal tissue regulation[18–20]. In addition, previous analyses showed distinct expression of ECM genes among sub-populations of epithelial stem cells[14,15,21]. These different ECM components may serve to anchor-specific stem cells in the niche, and be involved in communication between epithelial stem cells and adjacent dermal cell populations. Indeed, BM proteins derived from bulge epithelial stem cells provide a niche for arrector pili muscles and mechanosensory nerve complexes[14,15]. Similarly, type IV collagen, laminin and proteoglycans were detected in the BM at HG–DP interface or within DP[22]. One laminin α chain, laminin α5 (Lama5), is present in the BM of developing HGs and required for HF morphogenesis[23]. Although the BM at the HG–DP interface could be critical in the HF regeneration cycle, its molecular identities and functions, as for other regions of the HF, remain largely unknown.

Here, we systematically and semi-quantitatively identify the cellular origins, molecular identities and tissue distribution patterns of ECMs in the mouse HF at high spatial resolution. Our study provides the comprehensive overview of the ECM landscape within the adult HF and highlights how BM composition and structure are exquisitely tailored for individual inter-tissue interactions. Our study further reveals remarkable molecular complexity and spatial specialization of BMs in the HG–DP interface, which is involved in HF regeneration and HG–DP anchoring.

## Results

**Global ECM gene expression profiling in adult mouse HFs.** Deeper sequencing is required to obtain comprehensive genome-wide ECM gene expression profiles, including for low-abundance genes. Thus, we pooled different epithelial and dermal cell populations from adult telogen dorsal skin using cell sorting. We purified the following basal epithelial stem/progenitor cells (integrin α6+), resident in the lower isthmus (LI) (Lgr6+), UB (Gli1+), MB (CD34+), HG (Cdh3+) and unfractionated basal epithelial stem/progenitor cells (basal; mostly from the interfollicular epidermis (IFE)) from Lgr6-GFP-ires-CreERT2, Gli1-eGFP, Cdh3-eGFP and wild-type mice (Fig. 1a–e and Supplementary Fig. 1a–e) using our protocol[14]. Two dermal cell populations, DP cells (Lef1+/Pdgfra+) and pan-dermal fibroblasts (pan-DF, Pdgfra+), were also isolated from Lef1-eGFP and Pdgfra-H2B-eGFP mice (Fig. 1a, f, g and Supplementary Fig. 1f–l). After verifying purity by qRT-PCR on the HF region-specific genes[14] (Fig. 1h), each isolated population underwent RNA-seq. Principal component analysis (PCA) and hierarchical clustering showed that all biological replicates clustered together and were significantly different from other samples (Supplementary Fig. 1m, n).

To investigate the global ECM gene expression correlations among these cell populations, we performed Spearman's rank correlation coefficient analysis with all expressed genes, all 281 annotated ECM genes, called the 'matrisome' (see 'Methods'), and non-matrisome genes in distinct cell populations[6] (see Supplementary Table 1 for the list of matrisome genes and Supplementary Fig. 2 for their gene expression patterns). Upon using all expressed genes, we observed strong correlations among epithelial cell populations and dermal cell populations, respectively (Fig. 1i). However, when matrisome genes were used, DP showed a stronger correlation with HG ($0.842 \pm 0.018$) than pan-DF ($0.803 \pm 0.018$), even though HG cells are keratinocytes and DP cells are fibroblasts. Upon using non-matrisome genes, the HG–DP block became unclear, partly due to the lower correlation between other epithelial cell populations and dermal cell populations in matrisome genes. We further divided matrisome genes into 67 BM genes and 214 interstitial ECM genes, showing that interstitial ECM genes contribute to the stronger correlation between HG and DP ECM expression profiles (r of HG–DP and DP–pan-DF were $0.881 \pm 0.028$ and $0.793 \pm 0.021$, respectively). This demonstrates that the ECM expression profile of HG cells, especially that of interstitial ECMs, resembles not only those of other epithelial populations, but also that of DP. From another perspective, the ECM profile of DP resembles that of HG cells rather than that of pan-DF. Thus, ECM expression profiles of epithelial stem/progenitor compartments may be coupled with that of adjacent tissues to cooperatively establish extracellular microenvironments for local inter-tissue interactions.

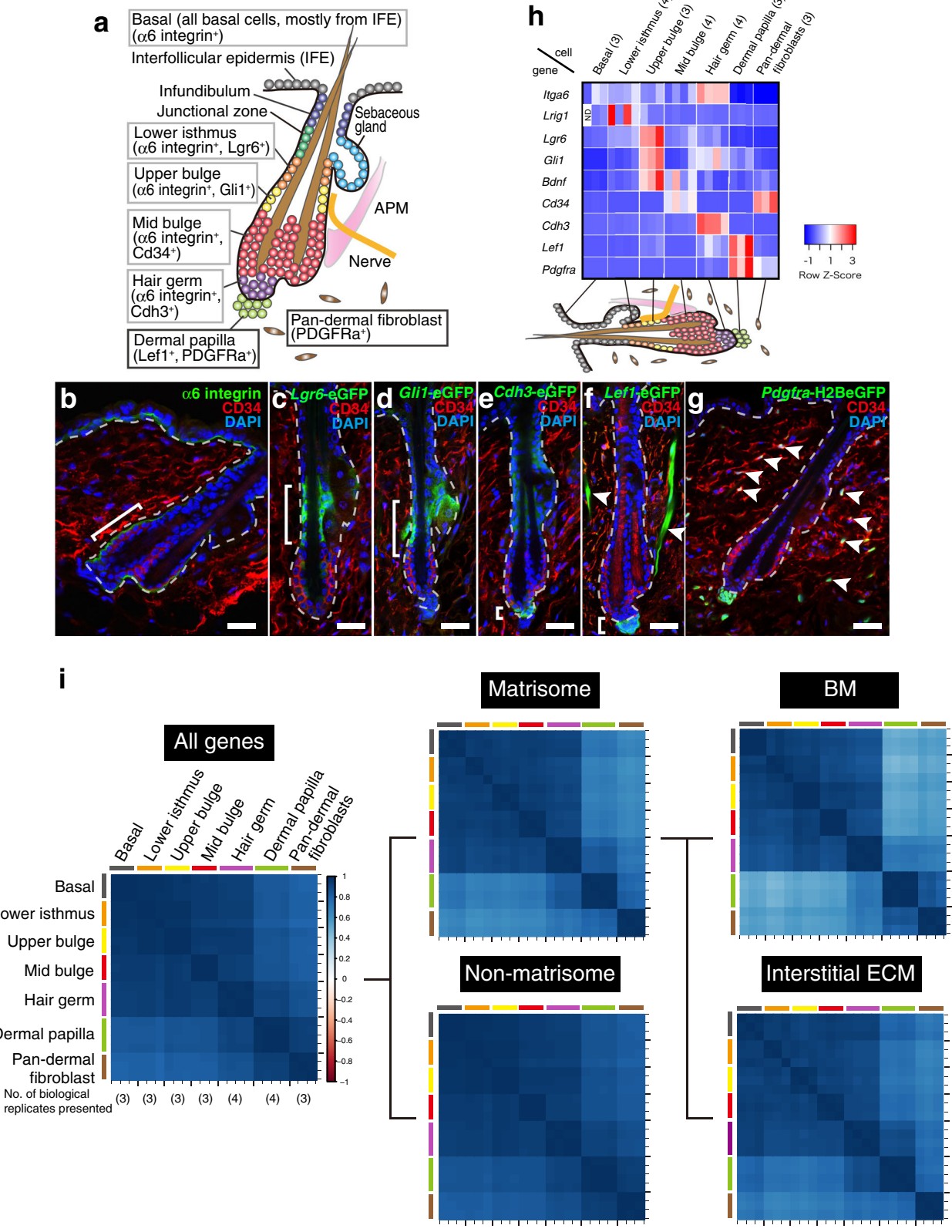

**Cellular origin of BM molecules.** Epithelial cells are generally considered to be the major source of the epithelial BM, but the cellular origin of BM components is not comprehensively understood. To gain insight into this, we quantitatively compared the expression levels of BM genes between basal epithelial cell (Basal) and pan-DF populations using normalized counts. Twenty-one BM genes were significantly highly expressed in the Basal (*p* adj < 0.05 and fold change > 4, Fig. 2). These genes can be classified into two categories: those encoding molecules functioning on the epithelium (*Lama3*, *Lama5*, *Lamb3*, *Lamc2*,

**Fig. 1 Targeted cell isolation and transcriptional profiling of mouse hair follicle. a** Graphical illustration of adult telogen HF compartments. Grey and black frames indicate epithelial and dermal compartments targeted in this study, respectively. IFE interfollicular epidermis. APM arrector pili muscle. **b–g** Tissue distribution of the markers for each HF compartment. Brackets indicate the target cell compartment in each panel. HF mid-bulge basal stem cells were labelled with α6 integrin (green) and CD34 (red) (**b**). Lower isthmus epithelial basal stem cells were visualized by *Lgr6*-eGFP (green) (**c**). Upper bulge epithelial basal stem cells were visualized by *Gli1*-eGFP (green) (**d**). HG cells were visualized by *Cdh3*-eGFP (green) (**e**). Dermal papilla (DP) cells and arrector pili muscles (arrowheads) were visualized by *Lef1*-eGFP (green) (**f**). Dermal fibroblasts (arrowheads), including DP cells, were visualized by *Pdgfra*-H2BeGFP (green) (**g**). White dashed lines indicate epithelial–dermal borders. Scale bar, 30 μm. **h** Relative mRNA expression levels of HF region-specific genes in different sorted cell populations. mRNA levels are expressed relative to *Gapdh* and represented by *Z*-score values. Numbers of biological replicates for each sample group are indicated in the sample names. **i** Global ECM gene expression correlation among HF cell populations examined by Spearman's rank correlation coefficient analysis. Each cell represents overall transcriptomic expression similarity between a pair of samples with a range from −1 to 1. All genes, matrisome genes, non-matrisome genes, basement membrane (BM) genes and interstitial ECM genes are used. Values of each biological replicate are presented in these heatmaps. Numbers of biological replicates are indicated in the 'All genes' panel.

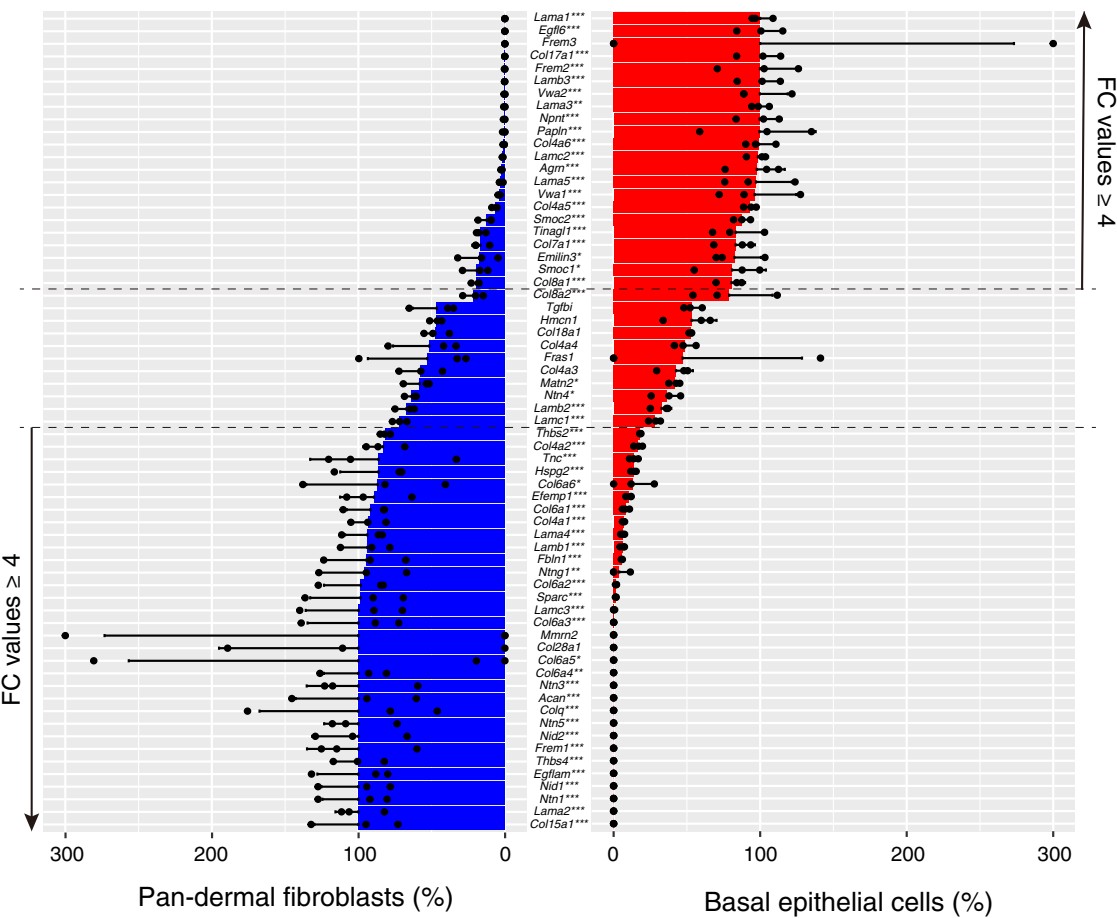

**Fig. 2 Comparison of basement membrane gene expression between epithelial and fibroblast cells.** Bar graph represents the ratio of expression of basement membrane (BM) genes in the basal epithelial cell population and pan-dermal fibroblast cell population. Data are mean ± SD of three biological replicates. Adjusted *p* values from two-sided Wald test are shown: *\*p* < 0.05; *\*\*p* < 0.01; *\*\*\*p* < 0.001. FC fold change.

*Col17a1*—key molecules for keratinocyte adhesion)[24] and on the dermis (*Egfl6*, *Frem2*, nephronectin (*Npnt*)—key molecules for epithelial–dermal interactions)[14,15,25]. In contrast, 30 BM genes were significantly highly expressed in the pan-DF with previously mentioned criteria. These genes include core BM genes, *Col4a1*, *Col4a2*, *Nid1* and *Nid2*. Other notable ECM genes were *Lama2*, *Lama4* and *Col6* isoforms, which mainly function on mesenchymal cells, such as in nerves, muscles and blood vessels[24]. Our data indicate that major BM molecules for keratinocyte adhesion are provided by basal keratinocytes themselves, but dermal fibroblasts are another major source of BM molecules.

We also compared expression levels of interstitial ECM genes between Basal and pan-DF. Eighty-eight interstitial ECM genes were significantly highly expressed in the pan-DF under our

criteria (Supplementary Fig. 3). Products of these genes include major interstitial structural ECM molecules such as *Col1a1*, *Col1a2*, *Col3a1*, *Col5a1*, *Bgn*, *Dcn*, *Lum*, *Fn1*, *Igfbps* and *Spp1*. Interstitial proteoglycans were almost exclusively expressed in pan-DF. However, many interstitial matrix genes were also expressed by Basal. Their distinct expression patterns among different epithelial cells are described in Fig. 3.

**ECM genes differentially expressed in distinct epithelial stem cells.** To identify ECM genes differentially expressed among different epithelial stem/progenitor cell populations, we performed differential gene expression analysis with DESeq2. Significantly differentially expressed ECM genes were identified and further subjected to hierarchical clustering analysis (Fig. 3). We cut the

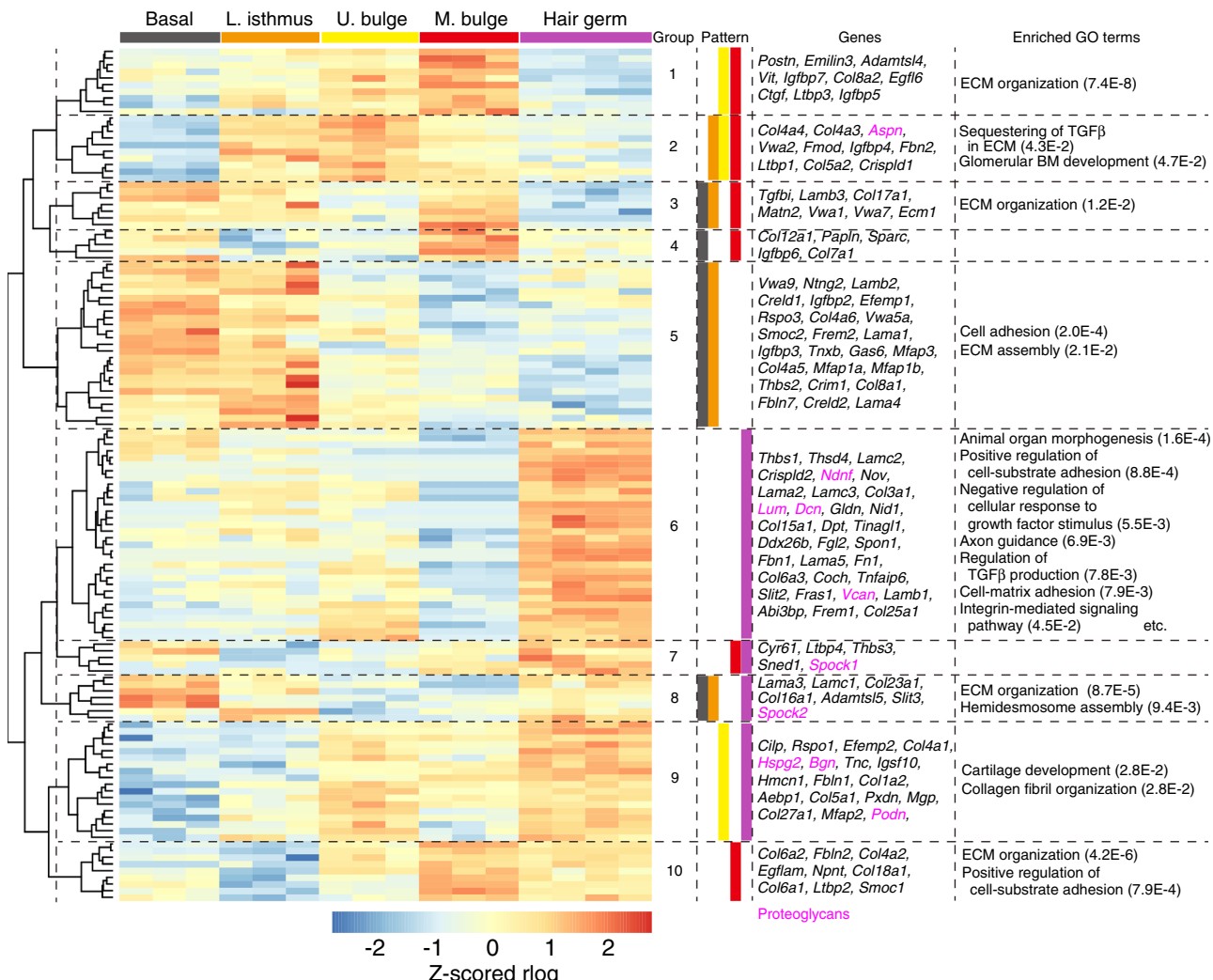

**Fig. 3 Hierarchical clustering of differentially expressed matrisome genes in epithelial sub-populations.** *Z*-scored expression values (rlog normalized values) of each biological replicate are indicated according to a heat gradient as shown at the bottom of the heatmap. Dendrogram tree was cut at the height where it gave ten sub-gene groups (epi-groups 1–10). To annotate ECM gene expression patterns in each epi-group, the epithelial regions were statistically divided into two groups by *k*-means clustering. The cell group expressing the cluster-composing ECM genes at a level higher than the other group was classified as epithelial regions showing relatively high expression of cluster-composing ECM genes. The expression patterns of each epi-group are colour encoded. Genes in magenta indicate proteoglycans. Enriched GO terms analyzed with region-specific ECM genes are shown with their FDR values.

dendrogram at the height where it gave ten sub-gene clusters, which we named epi-groups. In each epi-group, we determined epithelial regions where these grouped genes are relatively highly expressed using *k*-means clustering (*n* = 2, see 'Methods').

Our analysis identified nine MB-enriched ECM genes (epi-group 10). These genes play important roles in tendon and chondrocyte morphogenesis[26], suggesting the production of an ECM niche for interaction with muscles. Indeed, one MB ECM protein, *Npnt*, plays critical roles in arrector pili muscle development[14,15]. *Col17a1*, an important transmembrane ECM component for HF and IFE stem cell maintenance, was also identified as a Basal (IFE)/LI/MB-specific ECM gene (epi-group 3)[20,27]. Sorting these functionally important ECM genes into corresponding ECM gene groups demonstrates the reliability of the analysis.

Our analysis also identified 32 unique ECM genes highly enriched in HG (epi-group 6). The morphogenesis and growth factor-related Gene Ontology terms were overrepresented for these ECM genes (Fig. 3). For example, *Thbs1* (thrombospondin-1),

*Tnfaip6* (TSG-6) and *Spon1* (spondin-1/F-spondin) are involved in many morphological processes through TGF-β family signal regulation[28–30]. Consistent with these findings, dermal-derived TGF-β2 is critical for HG cell activation during the hair cycle[31]. Another key signalling pathway for HG–DP interactions is Wnt/ β-catenin signalling pathway[32]. *Rspo1* encoding Wnt agonist R-spondin-1[33], and *Bgn* and *Dcn*, encoding Wnt-related proteoglycans, were identified as HG-enriched ECM genes (epi-groups 6 and 9). Given that proteoglycans regulate the distribution and activity of soluble signalling molecules[4], we examined their expression patterns. Strikingly, nine out of ten proteoglycan genes in the heatmap were categorized into HG-enriched groups (epi-groups 6–9). These results demonstrate that HG cells express ECM genes involved in morphogen/growth factor regulation.

The expression patterns of ECM receptor genes were also examined. Hierarchical clustering revealed three major gene clusters—epithelial-type receptors, dermis-type receptors and common receptors (Supplementary Fig. 4a). Laminin-binding receptors, *Itga3*, *Itga6*, *Itgb4*, *Dag1* and *Bcam*, and their associated

tetraspanins were highly and broadly expressed in epithelial cell populations. Some of the dermis-type integrins, such as *Itga5*, *Itga9*, *Itgb1* and *Itgb3*, were more highly expressed in HG cells than in other epithelial cells (Supplementary Fig. 4a, b), suggesting the interactions of HG cells with interstitial ECMs. In comparison to ECM genes, ECM receptor genes showed broader expression patterns in the epithelium.

These results indicate that each epithelial stem/progenitor cell expresses region-specific ECM genes that play important roles in regional epithelial–dermal interactions, in addition to the ECM genes involved in epithelial–BM adhesion.

**ECM genes differentially expressed in different fibroblasts**. ECM genes differentially expressed in fibroblast populations were analyzed in the same manner as in the epithelium (Fig. 4a). Although DP cells are a subpopulation of dermal fibroblasts, they shut down the expression of many major interstitial ECM genes, including *Col1a1*, *Col1a2*, *Col3a1*, *Fn1*, *Sparc* and *Lum* (Fig. 4a). Instead, HG-specific ECM genes and BM genes were over-represented in DP cells, as confirmed by GSEA (Fig. 4a, b). Spondin family genes, including *Spon1*, *Rspo2* and *Rspo3*, were also upregulated in the DP, suggesting their roles in localized Wnt signal regulation.

**ECM protein tissue atlas of mouse HFs**. We further examined the tissue localization of regionally expressed epithelial and dermal ECM proteins by immunostaining and generated an ECM protein tissue atlas of mouse HFs. We used antibodies against 78 ECM proteins (tested 104 antibodies) and determined their immunostaining conditions with validation for specificity (Supplementary Data 1). Among them, 52 antibodies showed ECM-like extracellular deposition patterns. Immunostaining patterns of representative ECM proteins for several epi-groups are shown in Fig. 5a and those of all detected ECM components are shown in Supplementary Fig. 5. Protein deposition levels of 40 ECM proteins classified as epi-groups were first quantified and their relative expression levels were represented as a heatmap (Fig. 5b) and boxplots (Fig. 5d–h). Because some ECM proteins showed different deposition patterns around HG, deposition levels of the lateral HG (LHG) BM zone and the HG–DP interface BM zone were separately measured. The heatmap showed an overall correlation in tissue localization pattern of mRNAs and proteins of ECM molecules. This is supported by statistical tests for the differential protein deposition levels for each region-specific ECM gene category (Fig. 5d–h), suggesting that most ECM proteins are locally synthesized and deposited into matrices. However, we also identified ECM molecules showing clear discrepancies between mRNA and protein tissue localization patterns, including *Crim1*, *Col4a1*, *Tnc* and *Smoc1*, which might be caused by post-transcriptional mechanisms and secretion from other non-epithelial cell populations (see Supplementary Data 2).

We then examined the deposition of DP-specific ECM proteins in epithelial BM zones (Fig. 5c, i). We divided DP ECM genes into three groups based on their mRNA expression patterns in the epithelium—'DP only', 'DP and HG' and 'DP and non-HG epithelium'. Protein products of most DP-specific ECM molecules accumulate in HG BMs, either lateral or interface, indicating that HG BM zones not only comprise HG-derived ECMs, but are also compositely specialized by the ECM from HG and DP. The biological significance of this specialization is explored below.

**BM micro-niches along epithelial–mesenchymal interfaces**. We next probed the diversity within the BM niches in the HG–DP unit with the ECM protein tissue atlas. The first notable feature was the lack of reticular lamina components COL6 (*Col6a1*, *a3*,

*a6*) and COL7 (*Col7a1*) and hemidesmosome component COL17 (*Col17a1*) in the interface BM (Fig. 6a, b). The intracellular hemidesmosome component plectin was also absent there (Fig. 6b). Indeed, the number of electron-dense hemidesmosome-like structures was reduced at the interface BM (Fig. 6c–f). Notably, the lamina densa structure at the interface BM showed protrusions toward the dermis (Fig. 6g, arrows), which were absent in the lateral BM. These protrusions preferentially originated from hemidesmosome-like structures (Fig. 6g, arrowheads). The BM of the neuromuscular junction also lacks reticular lamina and extends protrusions from active zones to junctional folds of muscle fibre[34,35]. Thus, our analysis identified close parallels in molecular composition and structure of the BM between the HG–DP interface and neuromuscular junction.

We further identified two specialized BM structures in the HG–DP unit. Two core BM molecules, laminin α5 and perlecan, showed large protrusions from the interface BM into the DP centre where a nuclear signal was lacking (Fig. 7a). These protrusions resembled hooks that fasten DP to HG (Supplementary Fig. 6a). Super-resolution three-dimensional microscopy confirmed that these protrusions were continuously connected to the interface BM (Supplementary Video 1). Thus, we named this ECM structure the 'hook BM'. The hook BM also contains other major BM molecules, including laminin α2, α4, β1, β2, γ1, γ3, nidogen-1, -2 and COL4A1, but not laminin-332, suggesting that the major laminin isoforms in the hook BM are laminin α2, α4 and α5 chain-containing laminins (Fig. 7b). We also noticed a mesh-like deposition of perlecan within the DP, which was directly connected to the interface and hook BMs (Fig. 7a). We named this ECM structure the 'mesh BM'. The mesh BM also contains laminin α4, β1, γ1, nidogen-1, -2 and COL4A1, but not laminin α2, α5, β2, γ3 and laminin-332, suggesting that the major laminin isoform of the mesh BM is laminin-411. We also identified other BM molecules in the hook and mesh BMs (Fig. 7c and Supplementary Fig. 6b). Both epithelial and dermal cell compartments contribute to producing these components (Fig. 5). Taking these findings together, the ECM protein tissue atlas revealed exquisite molecular and structural diversity of BM micro-niches at the HG–DP interface (Fig. 7c) and showed that the BM is the primary ECM niche for DP cells.

To identify potential ECM receptors for these BMs, we examined the mRNA expression levels of major ECM receptors in DP and pan-DF. DP cells expressed higher levels of laminin-binding integrins (*Itga6* and *Itga3*)[36], Wnt signal regulator glycans (*Gpc1* and *Gpc2*)[37] and TGF-β activator *Itgb8*[38] than pan-DF (Supplementary Fig. 4a). Immunohistochemical detection of integrins showed that α3, α5, α6, α8, β1 and β4 were enriched at the interface BM, α6, αv and β1 were located at the hook BM, while α5, α6, α9, αv and β1 were enriched at the mesh BM (Supplementary Fig. 6c). Super-resolution microscopic images revealed that integrins α6 and β1, which form an α6β1 heterodimer that binds to laminin α5 chain-containing laminins[36], tightly associated with laminin α5 in the interface and hook BMs, suggesting that α6β1 on DP cells interacts with these BMs (Fig. 7d). Electron microscopy visualized the close associations of BM protrusions and cellular protrusions of DP and dermal stem cells at the interface and hook BMs (Supplementary Fig. 7a–e). Cell–cell interactions among DP and dermal stem cells were rarely observed; instead, the interface, hook and mesh BMs cohered these dermal cells. These results indicate that epithelial HG cells, DP cells and dermal stem cells aggregate by tangling with a continuous BM structure, exhibiting regionally specialized molecular compositions and structures.

Our findings revealed a remarkable degree of molecular and structural complexity of the BM niches and a variety of cell–BM interactions at the HG–DP interface. They also revealed

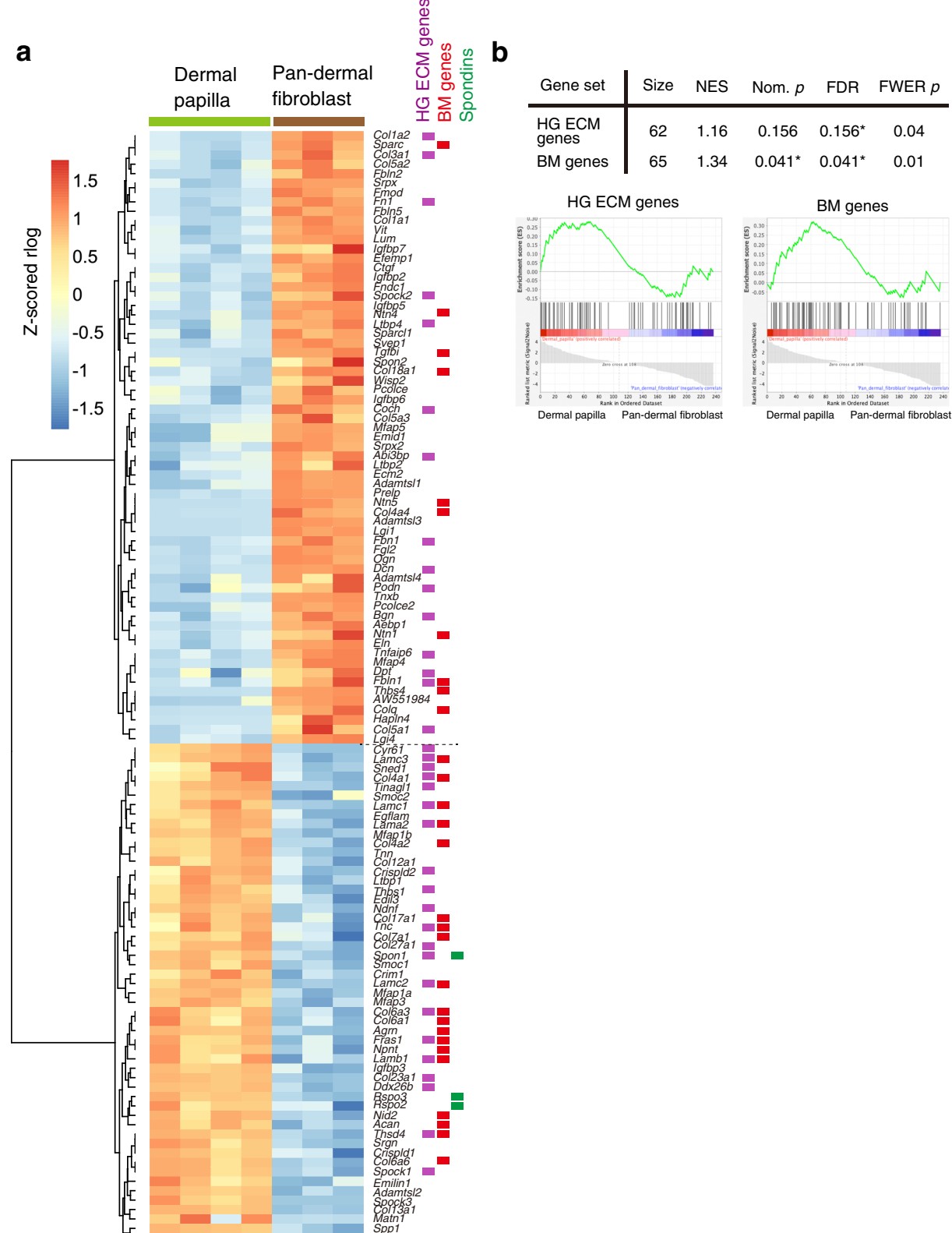

asymmetrically organized side-specific heterogeneity in BM composition and structure in this inter-tissue interface.

**Requirement of epithelial laminin α5 in hair cycle regulation.** Laminin α5 appeared to be a major cell adhesion ligand for both

HG and DP cells, while also reportedly being involved in HF morphogenesis[23]. Therefore, we examined the effects of *lama5* gene deletion on HF regeneration. To investigate the cellular origin of laminin α5, *Lama5* floxed mice were crossed with Keratin 5-Cre mice, which specifically express Cre in basal epithelium. In the mutant, laminin α5 immunoreactivity disappeared

**Fig. 4 Differential expression of matrisome genes between different fibroblasts. a** Heatmap showing hierarchical clustering of matrisome genes that are significantly differentially expressed between dermal papilla (DP) and pan-dermal fibroblasts (pan-DF). Z-scored expression values (rlog normalized values) of each biological replicate are indicated according to a heat gradient as shown at the bottom of the heatmap. ECM genes classified as HG-specific genes, basement membrane (BM) genes and Spondin family genes are indicated by colour bars next to each gene name. **b** Gene set enrichment analysis (GSEA) for HG ECM genes and basement membrane (BM) genes. Upper panel shows the results of GSEA for hair germ (HG) ECM genes and BM genes against an expression data sets of DP and pan-DF. ECM genes differentially expressed between DP and pan-DF are used. Lower panels show GSEA plots. Asterisks indicate statistical significance (<25% in FDR and <5% in nominal *p* value). NES normalized enrichment score, Nom. *p* nominal *p* value, FDR false discovery rate, FWER *p* family-wise error rate *p* value.

from the interface and hook BMs (Fig. 8a). Accumulation of integrins α6 and β1 in the hook and interface BMs was also abolished (Fig. 8b). Thus, epithelial-derived laminin α5 is the major ligand of these integrins in both HG and DP cells.

*Lama5* newborn cKO mice displayed reduced hair growth with defects in the tissue geometry of HG–DP interface (Supplementary Fig. 8a, b). We investigated the continuum of hair cycle stages by observing hair cycle domain patterns on living mice[39]. In the mutants, the onset of the first catagen was delayed, while HFs entered the next anagen at the same timing as the control mice without showing clear telogen transition and entered the catagen phase earlier than the control (Fig. 8c, d). In the second telogen phase (~P45–80), the mutants showed precocious anagen entry and exhibited a tail-to-head hair regenerative wave in dorsal skin (Fig. 8d, e), suggesting misregulation of signalling events for cyclic HF regeneration.

TGF-β/SMAD2 signalling pathway is a critical regulator of hair cycle progression[31]. It is activated in the HG in response to DP-derived TGF-β several days before the proliferative activity is seen within the HG. In early telogen phase (P49) of control mice, only a small fraction of HG cells in both neck and hip skin regions was positive for pSMAD2, while in the mutants, pSMAD2 signal was greatly increased in the HG in both skin regions (Fig. 8f and Supplementary Fig. 8c). HFs in the mutant hip region showed early anagen morphology, while those in the neck region still exhibited telogen HF morphology. Given that HFs in the mutant neck region enter the anagen phase around P63–67 (Fig. 8e), the strong pSMAD2 signal in mutant neck P49 HFs reveals that mutant HFs activate SMAD2 well before anagen entry. Thus, our results suggest that TGF-β signalling in the HG of mutant HFs is highly activated from early telogen phases.

Abnormal early anagen and catagen entry phenotype is also observed upon DP-specific β-catenin gene deletion[40]. Thus, we investigated the change in DP cell identity by examining the expression of LEF1, a key DP marker mediating canonical Wnt signalling[41] and HF development and regeneration[42,43]. Nuclear LEF1 was detected in DP cells in the telogen HFs of control mice (P84), but not in those of *Lama5* cKO mice (Fig. 8g). Meanwhile, HG cells showed similar levels of LEF1 expression. This implies that epithelium-derived laminin α5 is required for DP cell identity. Thus, these abnormal signalling states of HG and DP cells at least partly underlie dysregulated hair cycle progression in *Lama5* cKO mice.

**Involvement of epithelial laminin α5 in HG–DP anchoring.** The appearance of hook BM suggests that it functions as a hook to anchor the DP to the HG. Among hook BM proteins, laminin α5 and COL13A1 are implicated in the inter-tissue interactions in neuromuscular junctions[44,45]. COL13A1 is a transmembrane collagen expressed mainly by DP (Figs. 4 and 5 and Supplementary Figs. 2, 3, and 5). We examined the tissue localization dynamics of these hook BM components and β1 integrin during HF regeneration cycles. In the first telogen phase (P21), laminin α5 and COL13A1 were highly accumulated at the hook and interface BMs, while in the mid-anagen (P28, anagen IIIb), when

the DP was fully enclosed by the follicle epithelium, the deposition levels of these BM proteins were significantly decreased (Fig. 9a). However, just before catagen entry, their deposition levels were again increased and they formed hook-like structures in regressing HFs. β1 integrin was tightly associated with the hook and interface BMs. These results demonstrate that hook BM components emerge and increase only when HG–DP interface decreases in size from late catagen to telogen, suggesting the functional importance of hook BMs and their interaction with integrins in anchoring the HG and DP.

We further noticed that the mutant DP in *Lama5* cKO mice is smaller than that of control mice and some of these DPs were detached from the HG in second late telogen HFs (P84) (Fig. 9b). Thus, we measured the volume of DPs and their connectivity to the HG by detecting the tissue boundary of the DP and HG with perlecan staining because the mutant DP tended to have ectopically deposited melanin that sometimes hindered visualization of some DP nuclei. We found that mutant DP was smaller ($2624 \pm 1604\ \mu m^3$) than that of control HFs ($4063 \pm 1259\ \mu m^3$) (Fig. 9c). Violin plot suggested that detached or dispersed DPs in the mutant HFs are the major cause of shrunken DP (Fig. 9b, c). No HFs in control mice showed this phenotype. Thus, we conclude that epithelial-derived laminin α5 acts in anchoring the DP to the HG.

## Discussion

One important challenge in ECM biology is to understand how the ECM composition and structure are spatiotemporally specialized. However, the molecular landscape of the ECM composition and its pattern-forming processes in any organ remain largely unknown. Here, we systematically and semi-quantitatively identified the cellular origins, molecular identities and tissue distribution patterns of ECMs in the mouse HF at high spatial resolution. Our study provides the comprehensive overview of the ECM landscape within the adult HF and highlights how ECM composition is regionally specialized for each cell type and distinct inter-tissue interactions.

There are two major strategies to make regionally specialized ECM compositions: local synthesis at target cells/tissues or selective accumulation of ECM components from a distant source[3]. Our combinatorial systematic ECM mRNA and protein mapping approach revealed that, at least in the epithelial–dermal interface, most ECM molecules are synthesized locally and accumulated in adjacent ECMs, indicating that localized ECM expression is a major determinant of the ECM environment. This seems reasonable, but is also surprising because (1) there is no simple relationship between the in vivo concentration of a transcript and the extracellular protein from a particular locus[46], and (2) very different combinations of ECM proteins can be deposited locally together, despite the interaction and assembly of ECM molecules being considered to be regulated by specific molecular interactions[24]. This suggests the existence of molecular interaction networks within the locally expressed ECM molecules for their effective assembly and turnover. Our data also suggest that the ECM niche of each cell type can be well predicted from gene

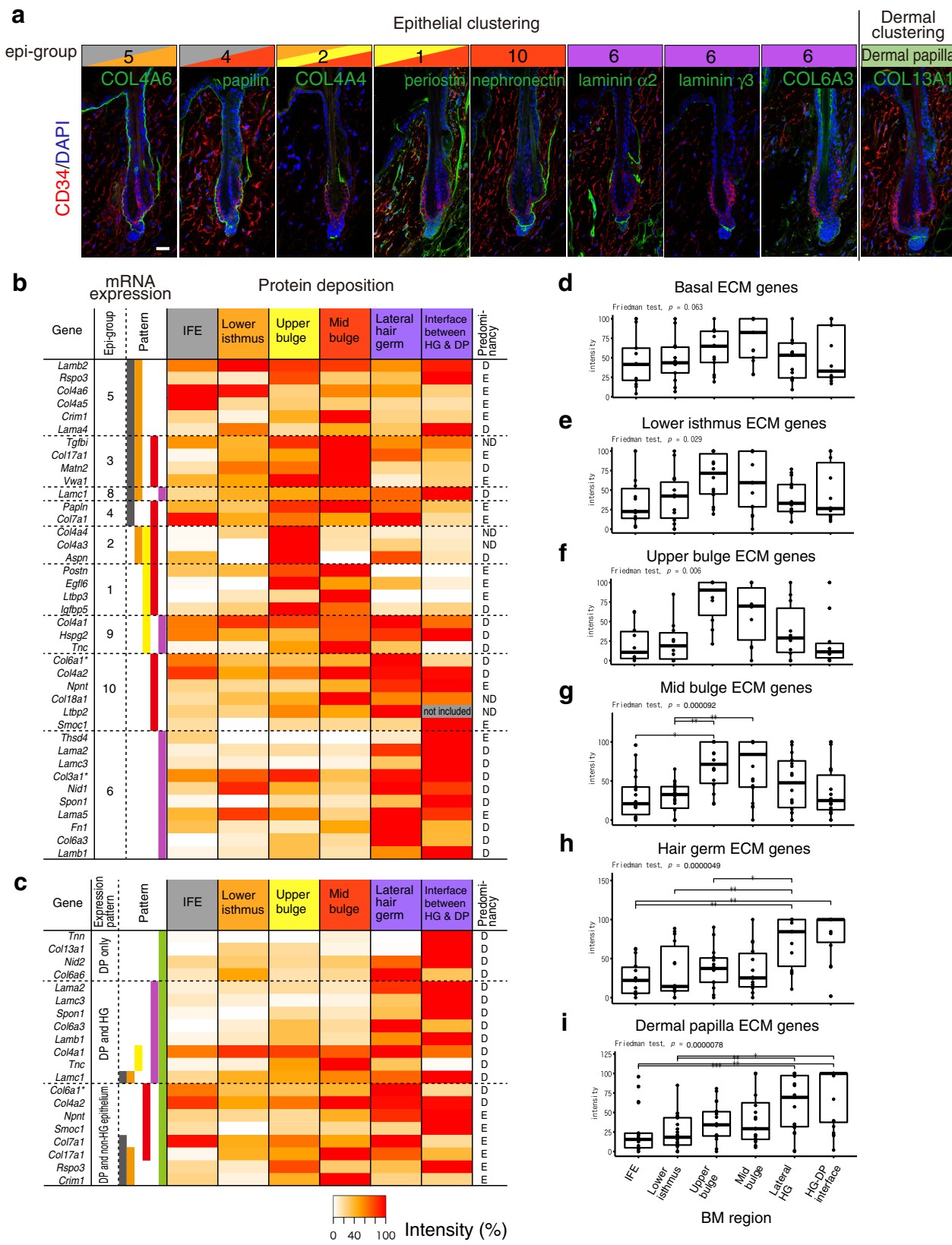

expression profiles of their own and neighbouring cells. This lays the foundation for predicting ECM niches of single cells within tissues using large-scale single-cell gene expression data sets, such as the Human Cell Atlas[47] and the mouse atlas *Tabula Muris*[48].

Our analysis also identified ECM proteins that exhibited inconsistent tissue localization patterns between mRNA and protein. This could be due to the biological complexity in regulating extracellular protein levels in certain tissue locations and

**Fig. 5 Specialized ECM niches and their cellular origins in the hair follicle. a** Representative protein tissue localization patterns of region-specific ECM genes. Immunofluorescence detection of each target ECM protein (green) and CD34 (red) in dorsal telogen HFs is shown with DAPI counterstaining (blue). Scale bar, 20 μm. **b** Heatmap displaying the quantified deposition levels of epi-group ECM proteins. Using immunofluorescent histochemical staining data (Fig. 5a and Supplementary Fig. 5), signal intensities of ECM proteins deposited at the divided basement membrane (BM) regions were measured as percentage values and depicted in a heatmap. Skin BM was regionalized into six areas as shown at the top of the heatmap. Epi-group numbers of each ECM gene and expression patterns assigned in Fig. 3 are indicated at the left panel of the heatmap. Major expression sources (comparison between basal and pan-dermal fibroblasts performed in Fig. 2 and Supplementary Fig. 3) are indicated at the right side. Asterisks on the gene symbol indicate the use of an antibody that does not distinguish a subunit composing ECM protein complexes. IFE interfollicular epidermis. **c** Heatmap displaying the quantified deposition levels of ECM proteins highly expressed in dermal papilla (DP) in the dermis. Expression colour codes are combined with dermal and epithelial classifications. **d–i** Box plot analysis of protein tissue distributions of region-specific ECM genes. ECM genes highly expressed in distinct HF regions are collected and their relative protein levels are represented as boxplots where the middle line is the median, the lower and upper hinges correspond to the first and third quartiles, respectively, the whiskers indicate 1.5 interquartile range. Statistical analysis was performed using Friedman test followed by two-sided Wilcoxon test with Bonferroni correction (*$p < 0.05$, **$p < 0.01$, ***$p < 0.001$; $n = 13$ genes for (**d**), $n = 14$ genes for (**e**), $n = 11$ genes for (**f**), $n = 19$ genes for (**g**), $n = 14$ genes for (**h**), $n = 20$ genes for (**i**)).

---

methodological differences between RNA-seq and immunohistochemistry. Extracellular proteins' levels in situ are mainly determined by transcriptional and translational regulation of transcripts and post-translational regulation of the protein products in resident and sometimes remote cells[3,46]. Molecule-specific long-range ECM transport and assembly mechanisms have been reported in *Drosophila*[49]. ECM receptors and other interacting ECM proteins are involved in the tissue localization of ECM proteins. Methodologically, our FACS-based RNA-seq analysis does not capture the transcript information of every cell residing in the skin, limiting the comprehensiveness of cell coverage for transcriptome information. In contrast, immunohistochemistry visualizes the sum of extracellular ECM proteins from various neighbouring cells, but cannot trace the cellular origin of each protein. Thus, spatial quantification of both mRNA and proteins for ECM molecules and understanding their relationships are major challenges[50,51]. Deeper and/or more diverse systematic molecular profiling and computational analysis of the expression and localization of ECM molecules and receptors will help us understand how distinct ECM niches are generated.

Our antibody-based ECM mapping revealed complex subcellular ECM distributions, leading to identification of the hook and mesh BMs. These matrices form molecularly and structurally fine-tuned ECM niches at the HG–DP interface. This resolution of spatial mapping cannot be achieved by other current proteome approaches including mass spectrometry[50–52]. Thus, merged antibody-based spatial ECM protein and single-cell mRNA expression profiles can precisely relate ECM composition to the positions of cells and molecules, providing distinct ECM niche information and a common anatomical reference for normal, aged and pathological tissue structures.

The BM can simultaneously function as both a tissue insulator and glue, keeping different cell populations close together but with a clear tissue boundary[3]. Our analysis showed that the molecular composition and structure of the BM are specialized for distinct inter-tissue interactions. The mid-bulge BM is composed of ECM molecules related to tendon and chondrocyte morphogenesis for the interaction with arrector pili muscles[15]. These tendon and chondrocyte-related ECM molecules are derived from epithelial bulge stem cells, but not from chondrocytes or related cell types, indicating that bulge stem cells actively participate in cooperative formation of the niche for epithelial–muscle interactions. To this end, bulge stem cells may need to activate a transcriptional network for tendon and chondrocyte ECM expression. Indeed, *Sox9* and *Scx*, master transcription factors for chondrocytes and tenocytes, are highly expressed in bulge stem cells[15,53,54]. In contrast, HG cells express a very different set of ECM genes, including those related to growth factor signalling, such as SMAD/TGF-β and Wnt

signalling. These signalling pathways are critical regulators of HG–DP interactions and HF morphogenesis and regeneration[16]. This marked difference in ECM expression patterns in adjoining epithelial compartments reflects their different counterpart tissues for inter-tissue interactions. Because fibroblasts also show remarkable heterogeneity in gene expression and functions[55], single-cell-level spatiotemporal analysis of ECM gene expression in both epithelium and dermis highlights cooperative interactions between certain sub-populations of epithelial and fibroblast cells in BM formation.

The absence of reticular lamina may allow intimate inter-tissue interactions by providing laminin–integrin interactions at both epithelial and dermal sides of the BM. The BM at the HG–DP interface lacks reticular lamina components, COL6 and COL7. This BM extends many protrusions from epithelial hemidesmosome-like structures toward the DP and dermal stem cells and forms laminin–integrin complexes at the dermal side. An analogous BM structure is present in the neuromuscular junction, where the reticular lamina is excluded and the BM extends protrusions from synaptic active zones toward junctional folds[34]. Laminin–integrin interactions can be observed on both nerve and muscle sides of the BM and play critical roles in neuromuscular junctions[34,56]. Moreover, COL7 is absent from lung alveoli, blood vessels and kidneys, where different tissues are tightly integrated via the BM that places laminins at both its sides[1,2,57–59]. Thus, the absence of reticular lamina probably enables laminin placement on both sides of the BM.

Although HG and DP cells actively communicate, little is known about the identity of molecules in their extracellular space. Our study revealed that the molecular composition of BM niches in the HG–DP unit is exquisitely tailored at the cellular level, probably to allow coordinated multi-lineage interactions. HG–DP interactions have the following key features: (1) DP cells form a packed cluster even though they are scatter-prone fibroblasts, (2) DP cells constantly attach to the HG region even though HFs undergo dynamic tissue regeneration and (3) HG and DP cells actively exchange signals via soluble factors such as Wnts, BMPs, FGFs and TGF-βs[16]. The hook and mesh BMs could help provide feature (1) because the hook and mesh BMs cohere laminin-receptor-expressing DP cells. The interface, hook and mesh BM complex can potentially underpin features (2) and (3) because these BMs are physically connected and preferentially composed of different adhesion and soluble signalling-related ECM molecules. In fact, deletion of the epithelial-derived interface and hook BM molecule laminin α5 induces DP cell detachment from the HG and alters SMAD2 activity in the HG. Laminin α5 is involved in many morphological processes via regulating integrin-mediated cell adhesion and growth factor-mediated signalling events in skin and other organs[23,60,61]. Thus, laminin α5 could

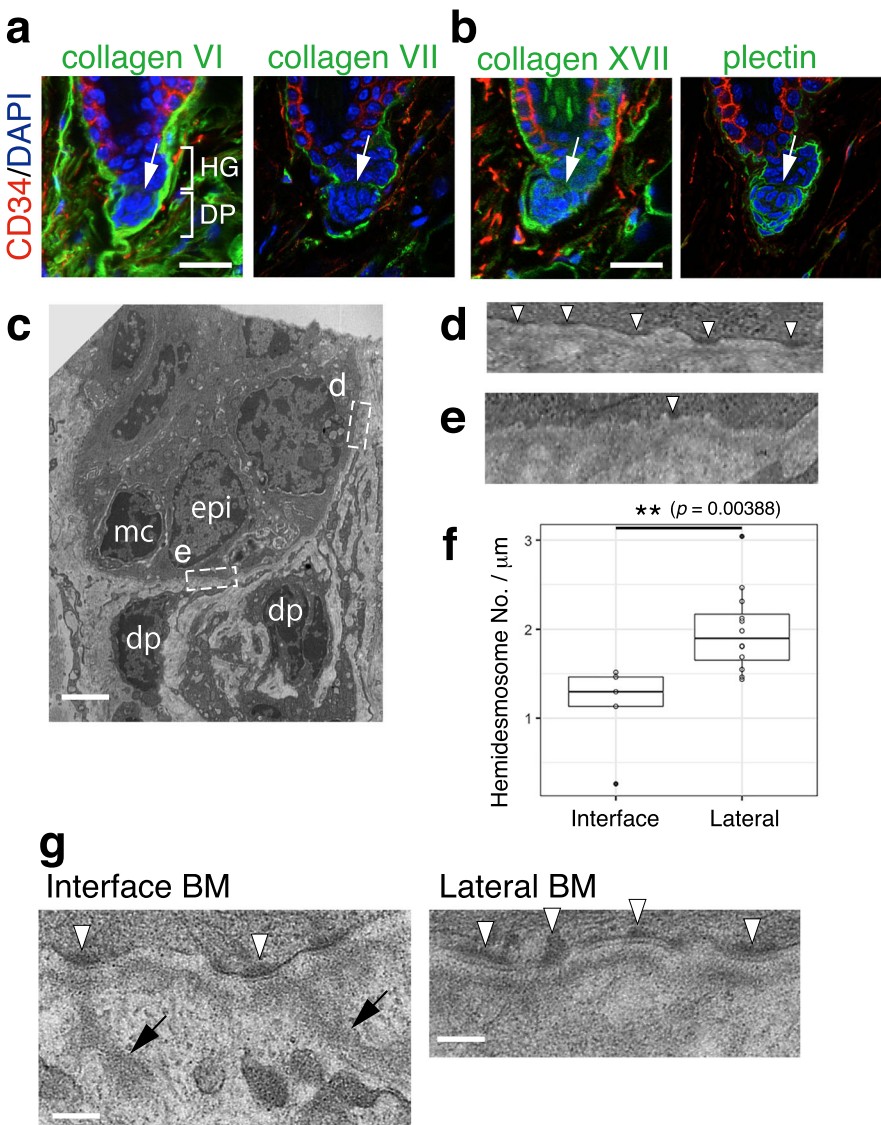

**Fig. 6 Unique molecular and structural properties of hair germ and dermal papilla interface.** Immunolocalizations of reticular lamina-related ECMs, COL6 and COL7 (**a**), and hemidesmosomal components, COL17 and plectin (**b**), in dorsal telogen HFs. These ECMs and plectin (green) were co-stained with CD34 (red) and DAPI (blue). White arrows indicate the interface between hair germ (HG) and dermal papilla (DP). **c** Transmission electron microscopy (TEM) image of HG and DP region. epi epidermal HG cell, mc melanocyte, dp DP cell. **d, e** Magnified images of HG–basement membrane (BM) adhesion sites located at the lateral side (**d**) and interface side (**e**) of the HG region (**c**). Hemidesmosome structures are indicated by white arrowheads. **f** Box plot of the hemidesmosome densities of HG cells located at the lateral or interface sides of the HG region. Data are indicated as boxplots where the middle line is the median, the lower and upper hinges correspond to the first and third quartiles, respectively, the whiskers indicate 1.5 interquartile range. **p < 0.01, Mann–Whitney U test (two tailed). n = 5 interface and 12 lateral regions examined over five different TEM images. **g** BM protrusions observed at the interface BM. BM protrusions extending into the interstitial space are marked with arrows. Hemidesmosome structures are indicated by white arrowheads. Scale bars: 20 μm (**a**), 2 μm (**c**), 500 nm (**g**).

function as a direct adhesion target for both HG and DP cells and regulate the tissue distribution and activity of growth factors.

Recently, Ge et al.[62] conducted single-cell RNA expression analysis of young and aged mouse skin epithelia and provided useful data to interrogate our findings. We analyzed their data set and confirmed that expression patterns of most differentially expressed ECM genes were consistent with our results, except for the absence of 15 HG-specific ECM genes (epi-group 6) in the list of differentially expressed genes created by the data set of Ge et al. (Supplementary Fig. 9a, b). This difference is potentially due to the cell population bias resulting from the difference in cell isolation protocol and RNA-seq procedure. Despite this difference, we compared ECM gene expression between young and aged cells

in the HG and bulge populations and found that ECM genes involved in the epithelial–mesenchymal interactions were listed as age-associated downregulated genes in the HG, including *Hmcn1* and *Fras1* (Supplementary Fig. 9c, d)[63,64]. In the bulge population, *Npnt* and *Egfl6*, BM genes critical for the interaction of bulge stem cells with arrector pili muscles and sensory nerves, were downregulated[14,15]. These results suggest that BM-mediated epithelial–mesenchymal interactions are altered in aged HFs.

Inter-tissue interactions are essential for the development, regeneration and functions of most organs. They have their own tailored BMs as structural and functional interfaces of inter-tissue interactions. Thus, future work should further characterize the molecular and structural properties of distinct BMs and their

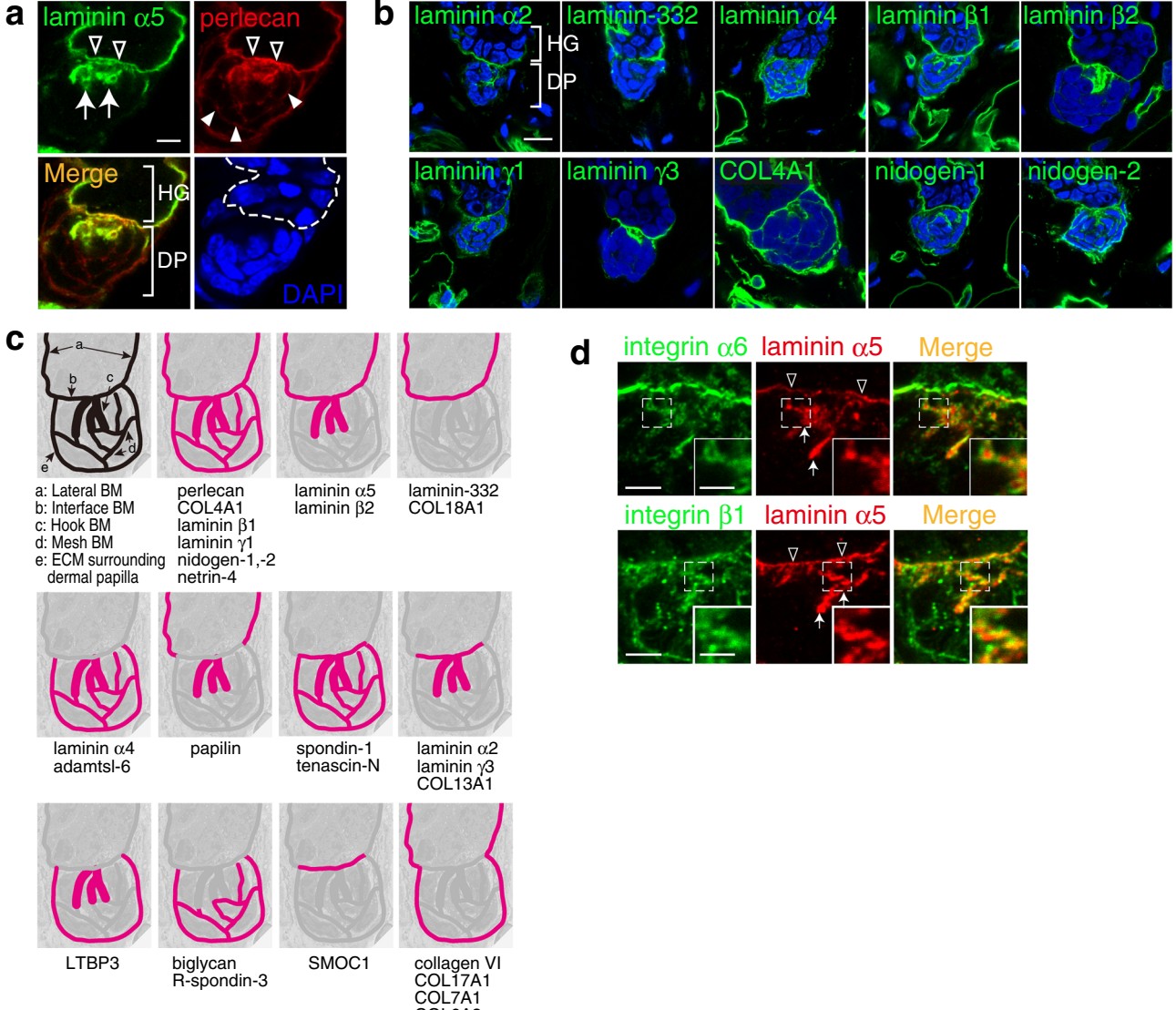

**Fig. 7 Identification of hook and mesh basement membranes. a** Immunofluorescence labelling of dorsal telogen HFs. Both laminin α5 (green) and perlecan (red) are detected in a hook-shaped basement membrane (BM; arrows) extending from the interface BM (open arrowheads). Perlecan also forms a mesh-like BM (filled arrowheads) within the dermal papilla (DP). Dashed line indicates the epidermal–dermal boundary. **b** Immunolocalizations of major BM proteins (green) in the hook and mesh BMs of dorsal telogen HFs. Nuclei were stained with DAPI (blue). **c** Graphical representation of the regional BM compositions in the hair germ (HG)–DP unit. Upper left-hand panel depicts the distinct BM structures. Other panels schematically summarize deposition patterns of BM components examined in (**a**, **b**) and Supplementary Fig. 5. **d** Close localization of dermal integrins with laminin α5-containing hook and interface BMs. Integrin α6 (upper panels, green) or β1 (lower panels, green) was co-immunostained with laminin α5 (red). Insets are magnified views of the dotted squares. Arrows and open arrowheads indicate the hook BM and interface BM, respectively. Scale bars: 5 μm (**a**, **b**), 3 μm (**d**), 1.5 μm (insets in (**d**)).

dynamics in inter-tissue interactions and reveal their significance in the coordination of multi-lineage interactions. This work provides a paradigm for understanding the role of BM heterogeneity in mediating inter-tissue interactions in multicellularity.

## Methods

**Mice**. *Lgr6-GFP-ires-CreERT2*, *Gli1-eGFP*, *Cdh3-eGFP*, *Lama5* floxed, *Lama5* knockout mice have been described previously[14,65]. *Lef1-eGFP* mice (STOCK Tg (Lef1-EGFP)IN75Gsat/Mmucd) were obtained from MMRRC. *Pdgfra-H2B-eGFP* mice (Pdgfra^tm11(EGFP)Sor)[66], K5-Cre mice[67] and R26-H2B-EGFP mice (CDB0238K)[68] were kindly provided by Dr. Philippe Soriano (Mount Sinai NY), Dr. Jose Jorcano (CIEMAT-CIBERER, Madrid, Spain) and Dr. Takaya Abe (RIKEN BDR), respectively. Mouse lines used for transcriptome analysis were backcrossed with C57BL/6N mice more than four times. R26-H2B-EGFP/*Lama5*^−/−^ mice were crossed with C57BL/6 albino mice several times to avoid possible imaging interference from melanin deposition. For cell sorting and immunohistochemical analysis, wild-type C57BL/6N mice were used. All animal experiments were

conducted and performed in accordance with approved Institutional Animal Care and Use Committee protocols in RIKEN Kobe Branch. Mice were housed in a 12-h light/12-h dark cycle and temperatures of 18–23 °C with 40–60% humidity.

**FACS**. Mouse dorsal epithelial cells were isolated as follows[14]. We utilized 8-week-old *Lgr6-GFP-ires-CreERT2*, *Gli1-eGFP*, *Cdh3-eGFP* and wild-type mice for epithelial cell isolation. The dermal adipose layer of dissected dorsal skin was scraped off with a scalpel. The skin tissue was treated with 0.25% trypsin solution (Nakalai tesque) at 37 °C for 1 h. The epithelial tissue was scraped off from the dermal tissue with a scalpel. For the sorting of LI (*Lgr6*-eGFP⁺), UB (*Gli1*-eGFP⁺) and MB (CD34⁺) epithelial stem cells, the separated epithelium was minced with scalpels and mixed with repeated pipetting to make a single-cell suspension. Cells were passed through a 40 μm cell strainer (Falcon, NC, USA). To deplete haematopoietic and endothelial cells (lineage-positive cells: Lin⁺ cells), the cell suspension was stained with PE-Cy7-conjugated antibodies for CD45 (eBioscience, CA, USA, 30-F11), TER119 (eBioscience, TER119) and CD31 (eBioscience, 390). To sort the target cells, the cell suspension was also stained with Sca-1-PerCP-Cy5.5 (eBioscience, D7), CD34-eFluor660 (eBioscience, RAM34), CD49f (integrin α6)-PE

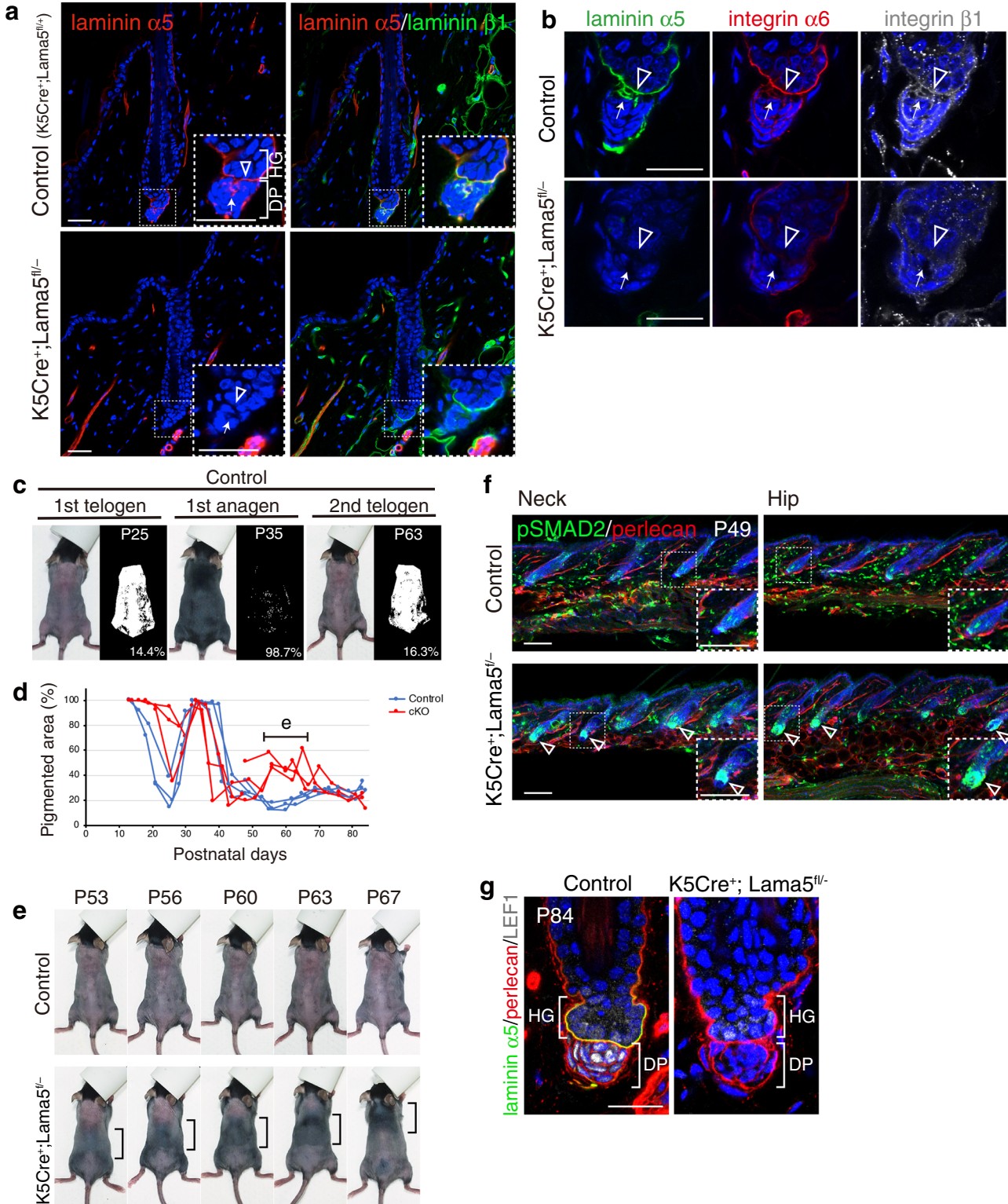

(eBioscience, GoH3). For the sorting of HG cells (*Cdh3*-eGFP+), the remaining dermal tissue were used because it retains the most HG cells[14]. The dermal tissue was minced with scalpels and incubated with 2 mg/ml of collagenase type I (Gibco, MD, USA) at 37 °C for 2 h with gentle mixing. Cells were passed through a 40 μm cell strainer. Single-cell suspension was obtained by repeated pipetting. The cell suspension was stained with the same antibodies used above and subjected to the sorting procedure with FACSAria II (BD Biosciences, CA, USA) according to the expression of cell surface markers, after gating out dead and Lin+ cells.

DP cells and pan-DF were isolated from the dorsal skin of 8-week-old female mice. The epithelial tissue was removed from the dermal tissue by scraping after trypsinization. The remaining dermal tissue was minced and treated with 2 mg/ml

collagenase type I at 37 °C for 2 h with gentle mixing. Single-cell suspension was obtained by repeated gentle pipetting and passed through a 40 μm cell strainer (Falcon, NC, USA). To eliminate Lin+ cells, the cell suspension was stained with the antibody cocktail used in the epithelial cell preparation. To further distinguish the target cell populations from the remaining epithelial cells, the expression of CD49f was examined using a PE-conjugated antibody. For the further analysis of dermal cell populations, the cell suspension was also stained with CD34-eFluor660 antibody. Cell isolation procedures are shown in Supplementary Fig. 1a–g. To determine the DP cell population, mRNA expression levels of *Pdgfra*, *Itga8* and *eGFP* were examined by qRT-PCR (Supplementary Fig. 1k, l). Cells were sorted with a FACSAria II according to the expression of reporter eGFP and cell surface

**Fig. 8 Epithelial-derived laminin α5 is required for the hair cycle regulation. a** Immunohistochemical examination of laminins α5 (red) and β1 (green) in the dorsal telogen HFs of control (*K5Cre+;Lama5fl/+*) and *K5Cre+;Lama5fl/−* mice. Magnified images of the dotted squares are shown (right panels). Open arrowheads indicate interface basement membranes (BMs). Arrows indicate hook BMs. **b** Immunohistochemical examination of laminin α5 (green) and integrins α6 (red) and β1 (white) in the dorsal telogen HFs of control (*K5Cre+;Lama5fl/+*) and *K5Cre+;Lama5fl/−* mice. Open arrowheads indicate interface BMs. Arrows indicate hook BMs. **c** Measurement of pigmented skin areas. Pigmented area, which is associated with the presence of anagen HFs, on the shaved dorsal skin was determined from binarized photos (right side of each panel) at various postnatal days (P). **d** Comparison of hair cycle patterns between control and mutant mice. Ratio of pigmented area represents the ratio of anagen HFs. Four litter pairs were examined. One pair of littermates was examined from P21 to P47 and another pair from P48. A horizontal bar indicates the period for detailed hair cycle pattern analysis shown in (**e**). **e** Precocious anagen entry and the formation of a travelling hair regeneration wave in *K5Cre+;Lama5fl/−* mice. **f** Immunofluorescence labelling of pSMAD2 (green), perlecan (red) and DAPI counterstain (blue) in the neck and hip skin regions of control (*K5Cre+;Lama5fl/+*) and *K5Cre+;Lama5fl/−* mice. Magnified images of the dotted squares are shown (right panels). Open arrowheads indicate strong pSMAD2 signals in the HG region of *K5Cre+;Lama5fl/−* mice. **g** Immunofluorescence labelling of laminin α5 (green), perlecan (red), lef1 (white) and DAPI counterstain (blue) in the dorsal telogen phase HFs (P84) of control (*K5Cre+;Lama5fl/+*) and *K5Cre+;Lama5fl/−* mice. Scale bars: 100 μm (**a**, **f**), 20 μm (**b**, **g**).

markers, after gating out dead and Lin+ cells. All the flow cytometry data were analyzed by FACSDiva (v7 and v8.0) and FlowJo (v9.9.3). We prepared three or four independent biological replicates and used them for qRT-PCR and RNA-seq.

**qRT-PCR**. qRT-PCR was performed as follows[14]. Total RNA was extracted from the sorted cells using an RNeasy micro kit (Qiagen). qRT-PCR was performed using Power SYBR Green PCR Master Mix (Life Technologies) with specific primer sets shown in Supplementary Table 2.

**RNA sequencing, mapping and expression quantification**. RNA sequencing and data processing were performed as follows[14]. Briefly, 10 ng total RNA samples extracted from FACS-isolated cells were subjected to library preparation using TruSeq Stranded mRNA Sample Prep Kit (Illumina) following the manufacturer's protocol with minor modification (shortened initial RNA fragmentation to 7 min). We generated three or four biologically independent cDNA libraries for each cell population. The prepared libraries were sequenced using the Rapid Run mode with 80 cycles on the HiSeq1500 (Illumina) followed by trimming low-quality bases and removal of adaptor sequences with Trim Galore (0.6.5). The processed reads were mapped to the mm10 mouse genome assembly using HISAT2 (2.1.0) with default parameter settings. To obtain a matrix of read counts, gene expression quantification was performed using the StringTie (2.0.4) programme. RNA sequencing and data processing described above were performed at the Laboratory for Phyloinformatics, BDR, RIKEN. Expression data for epithelial populations used in this study were reported in our previous study and deposited in BioProject (PRJNA342736)[14]. RNA-seq data obtained in this study have been submitted to the Sequence Read Archive as BioProject: PRJDB9477. RNA-seq read and mapping statistics for the analyzed libraries are summarized in Supplementary Table 3.

**Gene expression analysis**. To understand the expression patterns of ECM genes, we first compiled a list of ECM genes from the literature[9], and then defined 281 genes as our matrisome ECM genes (Supplementary Table 1). We further categorized the matrisome ECM genes into BM genes and interstitial ECM genes in a literature-based approach. To compare gene expression levels among the sorted cell compartments, the read count matrices were analyzed with DEseq2 (1.28.1) and other Bioconductor packages in R (4.0.2)[69]. Regularized log-transformation (rlog) value, size-normalized count value and gene length-normalized value (fragments per kilobase per million reads) were calculated by DESeq2 and used for data visualization. Charts of hierarchical clustering, expression correlation and PCA were plotted using ggplot2 (3.3.2) and corrplot (0.84). For hierarchical clustering, similarity was calculated using rlog counts by the *hclust* function with Spearman's rank correlation coefficient and the complete linkage clustering method. Spearman's rank correlation coefficient analysis was also used for expression correlation and PCA. Each ECM gene expression was visualized using the *heatmap.2* function.

To elucidate the regional expression of ECM genes, differentially expressed ECM genes were first identified by DESeq2. Wald test and likelihood ratio test were used for two-group comparison and multi-group comparison, respectively. Differentially expressed ECM genes (adjusted *p* value < 0.001) were clustered by hierarchical clustering. Gene Ontology terms enriched in each individual cluster were analyzed using the online PANTHER tool (http://pantherdb.org). Enriched biological process terms (over 40-fold enrichment) were evaluated for their *p* value and FDR. The epithelial regions were statistically divided into two groups in the individual sub-gene clusters based on the expression patterns of the clustered ECM genes by *k*-means clustering (*n_cluster* = 2). The cell group expressing the cluster-composing ECM genes at a level higher than the other group was classified as epithelial regions showing relatively high expression of cluster-composing ECM genes. To examine the expression ratio between basal epithelial cells (Basal) and pan-DF, normalized count values were calculated using the count function of DESeq2. Then, the ratio of Basal (*n* = 3) or pan-DF (*n* = 3) count to the total count was calculated. Gene set enrichment analysis was performed using normalized

counts of DP and pan-DF by GSEA (Broad Institute, version 4.1.0). Gene lists of HG-specific ECM genes and BM genes were manually generated and tested.

**Single-cell RNA-seq data analysis**. The raw sequencing data and data processing methods used were described by Ge et al.[62]. The raw data were obtained from Gene Expression Omnibus database (GEO) under accession code GSE124901, and were processed with the Seurat package (version 2.3.4) in R (version 4.0.2). The Seurat objects of young (2 months old) and old (24 months old) samples were constructed from GSM2558064 and GSM2558065, respectively. Quality control was performed as follows. For basic filtering, genes expressed in more than three cells and cells with at least 200 detected genes were kept. After filtering, 4173 and 4565 cells were retained in young and old samples, respectively. Global scaling was used to normalize counts across all cells in each sample (scale factor = 10,000). To integrate two data sets, common sources of variation between the two data sets were identified by canonical correlation analysis (CCA) using the top 1000 highly variable genes in each data set. Then, to remove the batch effect, the two data sets were aligned with CCA subspace using the first 20 CCs. To identify cell clusters, *FindClusters* function was used for CCA aligned data using the first 20 CCs. Identified clusters were visualized on a 2D map produced with tSNE (data not shown). We confirmed that clusters identified in our analysis showed almost the same top 10 conserved cluster markers as reported in the original paper. Finally, we defined distinct epithelial cell populations by the expression patterns of the lineage-specific markers used by Ge et al.[62].

To perform hierarchical clustering of expression patterns of matrisome genes across epithelial cell populations, the average expression of each gene was calculated using the *AverageExpression* function. The clustering results were visualized as a heatmap using the *heatmap.2* function.

For differential expression tests of single-cell transcriptomic data, MAST (Model-based Analysis for Single-cell Transcriptomics) was used[70]. The HG-enriched ECM genes (epi-groups 6–9 in Fig. 3) were used for the comparison between 'hair germ_young' and 'hair germ_old' data sets. Likewise, the MB-enriched ECM genes (epi-groups 1–4, 7 and 10) were used for the comparison between 'bulge_young' and 'bulge_old' data sets. Differentially expressed genes were defined as genes with adjusted *p* value < 0.05. Volcano plots were generated using the *ggplot2* R package.

**Antibody production**. To obtain a specific antibody against CRIM1 protein, a Japanese White rabbit was immunized with the recombinant extracellular region of CRIM1 protein and raised serum was collected. In detail, a cDNA fragment encoding the extracellular region of mouse *Crim1* (Leu35–Asp939) was amplified using cDNA derived from E16.5 mouse embryos with the following restriction enzyme site-tagged primer set: forward, GCGGCCCAGCCGGCCCTGGTCTGC CTGCCCTGTG, and reverse, CTCCTCGAGAGAGTCCAGTGATGAGTCTTC. Amplified cDNA was subcloned into the Sfi I-Xho I site of the pSecTag2A mammalian expression vector (Invitrogen). The CRIM1 extracellular region was transiently expressed and secreted by 293F cells using ExpiFectamine 293 (Gibco), and purified with a Ni-Sepharose 6 FF column (GE Healthcare, Little Chalfont, UK), following the manufacturer's protocol. Rabbits were immunized with the purified protein and high-titre serum was obtained (T.K. Craft Corp., Gunma, Japan). Antibody specificity was confirmed by immunostaining using mouse embryonic skin.

Rabbit antiserum to mouse laminin α5 was generated by immunizing rabbits with GST-fused I and II domains of laminin α5 (Lys2220–Leu2459). The I and II domains were amplified using cDNA derived from E16.5 mouse embryos with the following restriction enzyme site-tagged primer set: forward, CGGGATCCCGTA AACTCCGGGAGCCCACCGGGAC, and reverse, GGAATTCCTACTTGTCATCG TCGTCCTTGTAATCCAGGTGCTCTAGGTCCTCCTTAG. Amplified cDNA was subcloned into the EcoR I site of the pGEX-6P-1 expression vector (GE Healthcare). The antigen was expressed in BL21 and purified with a Glutathione Sepharose 4B column (GE Healthcare), following the manufacturer's protocol.

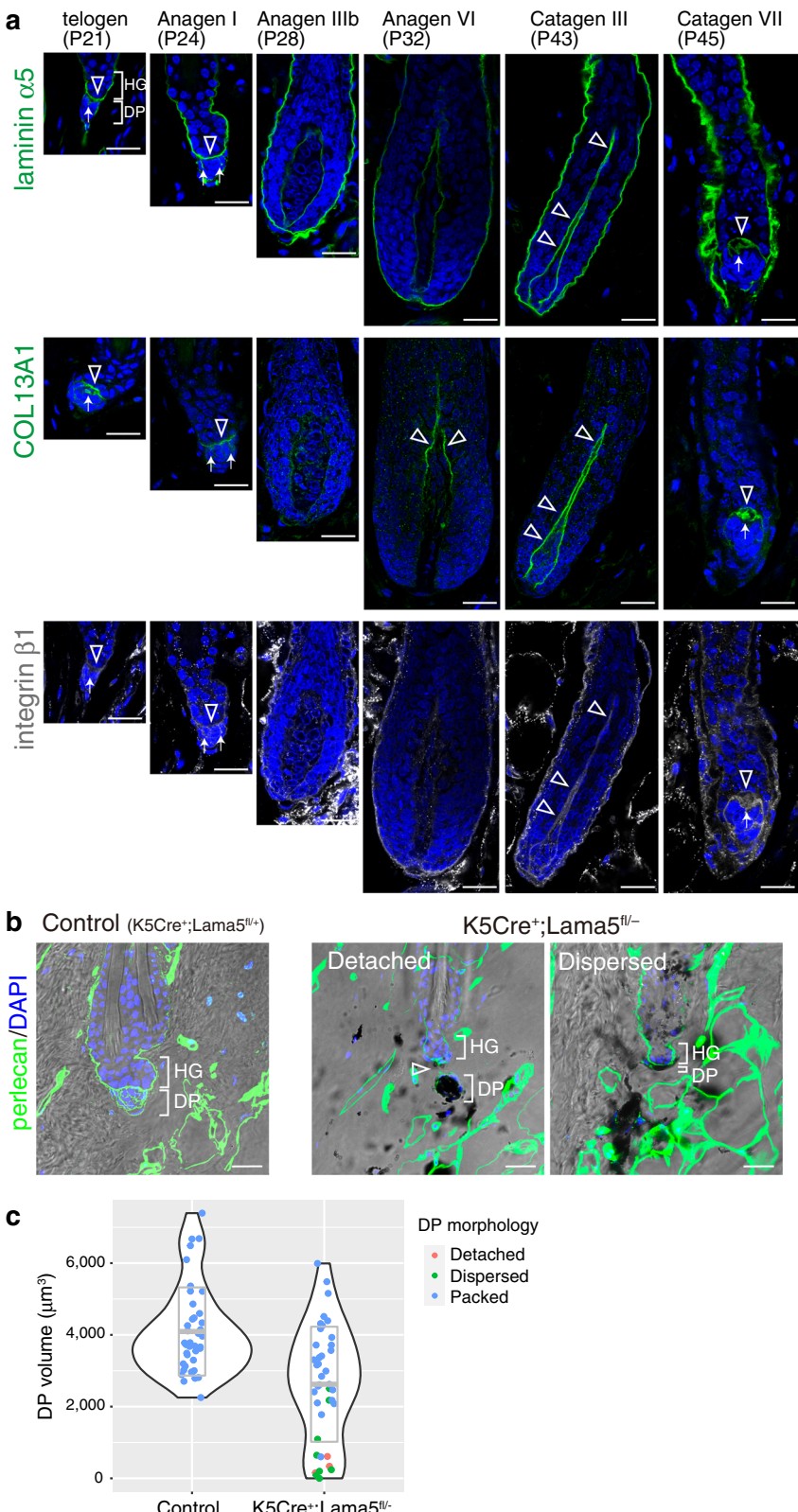

The antibody in the antiserum was affinity-purified with antigen-conjugated CNBr-activated Sepharose 4B. The specificity of the antibody to mouse laminin α5 chain was confirmed by the absence of antibody immunoreactivity to tissue samples from mice with *Lama5* conditional knockout.

**Antibodies**. Details of the antibodies used in this study are summarized in Supplementary Data 1.

**Immunohistochemistry and imaging**. Whole-mount immunostaining of mouse dorsal skin was performed as follows[15]. Briefly, mouse skin tissues were dissected and fixed with 4% paraformaldehyde (PFA)/PBS for 1 h at 4 °C, and embedded in OCT compound after washing with PBS. For acetone fixation, dissected skin was directly embedded in OCT compound. Skin sections (150 μm thick) were made using a cryostat (Leica, Wetzlar, Germany) and washed with PBS. Acetone fixation was performed by placing skin sections in −30 °C acetone for 15 min, followed by acid treatment with 0.1 N HCl/0.1 N KCl for 15 min after washing in PBS. Skin

**Fig. 9 Epithelial-derived laminin α5 is involved in hair germ–dermal papilla anchoring. a** Immunofluorescence labelling of laminin α5, COL13A1 and integrin β1 in the hair germ (HG)–dermal papilla (DP) interface region of different hair cycle stages of dorsal HFs. To avoid possible imaging interference from melanin deposition, C57BL/6 albino mice were used. Open arrowheads and arrows indicate the signals of laminin α5, COL13A1 and integrin β1 in the interface basement membranes (BMs) and the hook BM regions, respectively. **b** Immunofluorescence images of perlecan (green) and DAPI counterstain (blue) merged with bright field images of the HG–DP interface region of telogen phase HFs of control ($K5Cre^+;Lama5^{fl/+}$) and $K5Cre^+;Lama5^{fl/-}$ mice. Dark melanin pigments are visible. Open arrowheads indicate a gap between the HG and DP. **c** Quantification of DP volume in control ($K5Cre^+;Lama5^{fl/+}$) and $K5Cre^+;Lama5^{fl/-}$ mice. DP volume was three-dimensionally quantified using images taken in (**b**) and visualized by violin plot. Most of the DPs with a small volume (<2000 $\mu m^3$) are detached from the HG or dispersed. The middle line in the grey box indicates the mean, the lower and upper hinges correspond to the SD. $n = 40$ HFs from two mice. Scale bars: 20 $\mu$m (**a**, **b**).

sections were blocked with a blocking buffer (0.5% skim milk/0.25% fish skin gelatine/0.5% Triton X-100/PBS) for 1 h at 4 °C, and then incubated with primary antibodies diluted in blocking buffer overnight at 4 °C. Skin samples were washed with 0.2% Tween 20/PBS for 4 h and then incubated with secondary antibodies similarly to the primary antibodies. After that, skin samples were stained with DAPI, washed with 0.2% Tween 20/PBS for 4 h at 4 °C, and mounted with BABB clearing solution. Images were acquired using Leica TSC SP8 and Zeiss LSM880 with Airyscan (for 3D movie of the hook BM). Three-dimensional reconstructed images were produced using Imaris software (Bitplane, Oxford, UK).

**Antibody validation**. High-resolution confocal microscopy images of adult mouse skin tissues stained by indirect immunofluorescence were annotated for subcellular localization. Antibodies showing extracellular staining with ECM-like localization patterns were selected as validated ECM antibodies (Supplementary Data 1). We further annotated each antibody based on the tissue staining pattern around HF epithelium. These tissue staining patterns were compared with their mRNA expression patterns (for detail, see Image quantification). These results are summarized in Supplementary Data 1 and 2 and Fig. 5.

**Transmission electron microscopy**. Mouse dorsal skin tissues were dissected into 2–3 mm squares and immersed in fixation solution (2% PFA/2% glutaraldehyde/0.1 M phosphate buffer). The following steps were performed by Hanaichi Ultrastructure Research Institute (Okazaki, Japan). After washing with 0.1 M phosphate buffer, samples were post-fixed with 2% osmium tetroxide followed by step-dehydration with gradual substitution in higher-concentration ethanol (30, 50, 70, 90 and 100%) and finally 100% propylene oxide. Then, the samples were embedded in epoxy resin Epon812. Ultra-thin sections were cut, stained with uranyl acetate and lead citrate solution, and viewed with a JEM-1200EX (JEOL, Tokyo, Japan) transmission electron microscope at an accelerating voltage of 80 kV.

**Image quantification**. All quantification analyses were performed using Fiji software (ver. 2.0.0-rc-69). To calculate ECM protein intensities in the different BM regions, six epithelial regions (IFE, LI, UB, MB, LHG and interface region between HG and DP) were specified from HF morphology, CD34 staining pattern and their representative immunohistochemical patterns. The target BM regions were manually drawn in the colour split image, and their mean intensities were measured followed by subtracting the background intensity of adjoining epithelial regions. Nonspecific signals (e.g., hair shaft) were confirmed by immunostaining without primary antibodies and excluded from the measurement. Relative intensities were calculated as percentile values where the maximum-intensity region was 100. The data were represented as a heatmap chart by Bioconductor R. The accumulation tendency of regionally expressed ECM proteins was visualized as boxplots and statistically compared using Friedman's test followed by the Wilcoxon rank sum test with Bonferroni's correction. Pearson's correlation coefficients ($r$) between mRNA level and protein deposition of individual ECM components were calculated using normalized count data. Consistencies/discrepancies between mRNA and protein were judged using the following criteria: peak regions of both mRNA in the epidermal regions and protein deposition are the same, as well as $r$ value over 0.5. To quantify hemidesmosome-like structures and cellular protrusions, cellular perimeters facing the BM region or space of interest were measured by tracing freehand with a pen on the scale-set images. The lengths were used for calculation of the frequencies of appearance of hemidesmosome-like structures and cellular protrusions. Segmentation of the cellular fraction was manually performed using Fiji software. To quantify dorsal pigmented areas, binarized ROI images were first generated from the individual photos using Fiji software. Image-specific thresholds were determined manually between pigmented and non-pigmented areas from 8-bit greyscale images. Pixels corresponding to the pigmented area were counted using Fiji. A box plot graph was created using Bioconductor R.

**Statistics and reproducibility**. No statistical method was used to predetermine sample size. Statistical parameters including the numbers of samples and replicates, types of statistical analysis and statistical significance are indicated in the Results, Figures and Figure Legends. *$p < 0.05$; **$p < 0.01$; ***$p < 0.001$. For quantitative analysis, we have three or more biological replicates for each experiment, except for

Fig. 9c and Supplementary Fig. 1l, and reproduced with similar results. We show representative micrographs that came from at least two biological replicates.

**Reporting summary**. Further information on research design is available in the Nature Research Reporting Summary linked to this article.

## Data availability

All data that support the findings of this study are available within the paper and its Supplementary information files or are available from the corresponding author upon reasonable request. Previously published RNA-seq data[62] are available on GEO with accession number GSE124901. The RNA-seq data used and reported in this study are available in BioProject PRJNA342736 and PRJDB9477. Source data are provided with this paper.

## Code availability

Codes used in this study are available in GitHub: https://github.com/FujiwaraLab/Tsutsui_etal. https://doi.org/10.5281/zenodo.4620935.

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

## Acknowledgements

We thank Shigehiro Kuraku, Chiharu Tanegashima, Sean D. Keeley, Yuichiro Hara and Osamu Nishimura of the Laboratory for Phyloinformatics, RIKEN, for help with the RNA-seq and bioinformatics; Yasuko Tomono (Shigei Medical Research Institute) for antibodies against type IV collagens; Yoshiaki Hirako (Nagoya University) for antibody against type XVII collagen; Philippe Soriano (Mount Sinai NY) for *Pdgfra-H2B-eGFP* mice (Pdgfra^tm11(EGFP)Sor); Jose Jorcano (CIEMAT-CIBERER, Madrid, Spain) for K5-Cre mice; Takaya Abe (RIKEN BDR) for R26-H2B-EGFP mice; and RIKEN Kobe light microscopy and animal facilities for technical assistance. We also thank members of the Fujiwara laboratory for valuable reagents and discussions. This work was funded by RIKEN intramural funding, RIKEN 'Epigenome and Disease', JSPS KAKENHI (25122720, 26670533, 20H03706), JST CREST (JPMJCR1926), Uehara Memorial Foundation, Takeda Science Foundation and Cosmetology Research Foundation (all to H.F.). H.M. was supported by the RIKEN Junior Research Associate (JRA) programme.

## Author contributions

K.T. designed and carried out experiments, analyzed data and wrote the paper. H.M. designed and carried out experiments and analyzed data. K.A. supported statistical

analysis. A.N. provided technical support. R.M. examined phenotypes of whisker follicles of Lama5⁻/⁻ mice. J.H.M. provided Lama5 floxed and knockout mice. K.S. provided antibodies against vwa1, papilin, TGFBI, adamtsl-6, abi3bp, laminin β2 and γ3. H.F. conceived the project, designed and supervised experiments, analyzed data and wrote the paper.

## Competing interests

The authors declare no competing interests.
