## [Peer Review File · Nature Communications]

REVIEWER COMMENTS

Reviewer #1 (Remarks to the Author):

This manuscript provides new insights based on an exceptionally broad, comprehensive characterization of ECM proteins comprising the microenvironments at separate sites of hair follicles. The main contributions include introduction of novel, exciting concepts regarding hook-and-mesh organization of basement membranes in developing tissues, as well as cell-basement membrane interaction at the HG-DP interface. These findings might prove helpful in developing regeneration strategies. The contributions of this paper could become sufficient for Nature Communications if the documentation were to be made stronger in a revised manuscript. Specific concerns include the need to use more up-to-date RNA-seq analysis methods, clarification of methods and ruling out possible fluorescence artifacts, and clearer description of the Lama5-deletion mouse phenotype.

Major points

1. RNA-seq analysis methods: The authors should use more up-to-date methods because the use of outdated software can affect results and reproducibility. For example, Tophat and Cuffdiff not only have known issues but their use in bioinformatics has been overtaken by newer software such as HISAT2 and STAR (Dobin et al., Bioinformatics 2013). Once familiarized with their use, applying newer methods to re-analyze the data should not take much time and might resolve some of the discrepancies between RNA-seq and protein localization noted below.

A specific issue is that a FPKM-based normalization method is not ideal for differential expression analysis. RPKM/FPKM methods introduce a bias in the per-gene variances, particularly for genes expressed at low levels. Please use newer raw count-based methods like DESeq2 and EdgeR instead.
<https://bioconductor.org/packages/release/bioc/html/DESeq2.html>
<https://bioconductor.org/packages/release/bioc/html/edgeR.html>

2. Discrepancies between RNA expression and protein localization, particularly in Supplementary Fig. 3: The authors are to be commended for their comprehensiveness and completeness in showing immunohistochemistry staining along with mRNA localization, which provide valuable complementary resources. However, the discrepancies between some RNA-seq results and protein localization seem quite confusing. This confusion results in part from a lack of presentation of these discrepancies in the Results text and only in the Discussion, from how the images are presented (staining does not always conform to the description of what it's supposed to stain, and discussion of the significance of the discrepancies. Although their identification of multiple key markers at the mRNA and protein levels in which these two expression results agree is obviously valuable, this level of confusion for other putative markers is confusing.

Aside from the molecules that the authors mention in the Discussion to have inconsistent localization patterns, other proteins also show inconsistent staining patterns that should be mentioned. For example:

- Mid bulge-specific: Ltbp3 is positive in dermal papilla. Collagen VI stains along the upper bulge, mid-bulge, and hair germ regions.
- Hair germ-specific: Spondin-1 appears to be more dermal papilla-specific. No specific staining pattern of Decorin is evident.
- Upper bulge & hair germ: Fibronectin stains hair germ and dermal papilla and not upper bulge. No hair germ staining is shown with Peroxidase.
- Dermal papilla-specific: No obvious staining pattern is observed with Col6a5 except for the basal

layer.

Potential reasons for these discrepancies need further elaboration. The RNA-seq analysis methods as well as inherent methodological differences between RNA-seq and immunohistochemistry analyses may also account for this phenomenon.

However, some potential autofluorescence artifacts are evident in what appears to be hair shafts (e.g., Col17a1, Col13a1, Smoc1). Were autofluorescence controls used to rule out false positives for these samples?

3. With the Lama5 conditional knockout mouse model showing a non-essential phenotype for hair growth, Lama5 being required for “functional integrity of hair germ–dermal papilla interactions” is a vague possible overstatement. The wording should be more precise here.

4. The findings of mesh/hook BM, BM protrusions and cell-BM interaction at the HG-DP interface are quite interesting. More insights and, if possible, some additional speculation about their significance would be helpful.

Also, please add a recent paper (Ge...Fuchs et al., PNAS 2020) to your Discussion. They provide single-cell molecular characterization of murine hair follicles and show differences in ECM genes between young and old hair follicle stem cells. How do your findings compare to theirs?

Minor points

5. It is puzzling that the authors do not present a more-detailed hierarchical clustering analysis, e.g., in supplemental material, showing each of the separate biological replicates rather than just pooled data. Although the analysis shown in the main text is of course useful conceptually, this type of detailed hierarchical clustering analysis can provide readers with insight into the level of variability between different biological replicates.

6. Fig. 6g: Hemidesmosomes should be indicated by white arrows in the lateral BM as well, in order to compare with interface BM.

7. Please consider whether some detailed methods information presented in Results is needed. For example,

-spearman’s rank correlation coefficient analysis... (Line 113-114).

-“...based on the following criteria: a gene expression level in a cell population i) of FPKM value ≥ 3 and ii) more than 40% of the FPKM value of the most highly expressed cell population. Categorized genes were depicted using a Venn diagram... We identified a commonly expressed epidermal stem/progenitor ECM gene group at the middle of the Venn diagram” (Line 182-187).

8. Minor typographical errors:

-The first letter of mouse proteins should be capitalized.

-Sentence structure of headings should be consistent. Many headings are sentence fragments whereas others use complete sentence structure.

-The use of word “despite” was confusing at times.

-Line 104: add comma after (integrin $\alpha 6+$)

-Line 181: “cell-type-dependent”  “cell type-dependent”

-Line 201: “mid-bulge-“  “mid bulge-“

-Line 378: synthesis locally  local synthesis

Reviewer #2 (Remarks to the Author):

In the manuscript, Tsutsui et al addressed the molecular and spatial details of the basement membrane using quantitative transcriptomics and immunofluorescence. They used mouse telogen skin as a model to determine the expression profiles of ECM molecules from different epidermal/dermal compartments in homeostasis. This work provides a comprehensive picture of the cellular contributions from epithelial cells and fibroblasts, adding quantitative details to our understanding of the basement membrane. Specifically, the authors identified region-specific expression patterns, providing foundation for the niche-specific basement membrane architecture. To further their spatial understanding, the authors examined a total of 41 protein patterns via antibody immunostaining with outstanding clarity. They first interrogated the protein source by comparing protein with mRNA expression patterns, revealing mostly similarity that indicates local assembly as well as some discrepancy suggesting long-range transportations. Then they focused on the basement membrane between the epithelial hair germ and the mesenchymal DP, and identified the “hook” and “mesh” structures. Lastly, they showed the importance of one “hook” protein, laminin a5 to hair follicle structure and function. The data presented here are a beautiful combination of comprehensive transcriptome analysis and in-depth analysis of a regional BM spatial and functional details. These findings are of high importance and interest and will represent an important reference to the field moving forward. Below a number of suggestions for the authors to consider to improve coherence and/or accuracy where needed.

- The focus on basement membrane is obscured in the transcriptomic analysis, since the authors put much emphasis on interstitial ECM as well. Perhaps the interstitial ECM could serve as a control instead of a counterpart for the coherency of the story. For example, the authors used all matrisome for hierarchy clustering. It was not clear whether basement membrane matrisome only will have stronger/similar clustering. Additionally why pan-dermal fibroblasts don't cluster with basal epithelial cells: does that indicate distinct regulatory mechanisms involving basement membranes of different locations?

- Could the dataset in Figure 2 be more quantitative about the actual contributions from fibroblasts and epithelial cells? Would be helpful if the authors have a way to quantify the percentage of expressions between epithelial cells and fibroblasts

- It was not clear how the “rigorous validation for specificity” (line 241) for antibody staining is performed.

- The name “hook” is intriguing as it matches its appearance but may be leading to the assumption that the BM actually function as a hook to anchor DP to the hair germ. Additionally, could there be an opportunity to extract more information from their exciting laminin a5 knock-out model to understand the function of the “hook” BM? For example, were there any detachment of DP from hair germ in catagen for laminin a5 knock-out strains (if the hook is not working)? What is the cause of the hair cycle defect? Hair follicles appeared to have more DP cells in Lama5^{-/-}. Was that something consistent among mice and how do the authors explain this?

- The conclusion of continuous basement membrane (line 329) may not be always supported by some of the staining in figure 7. One can also imagine that two populations of BM in close proximity will also be revealed as connected in an immunostaining image, where the resolution is low. The captured EM data may provide some more opportunities to address this question?

Minor points:

- The introductions lack the context of telogen hair follicles. Adding references about the BM components and functional studies in DP may help a reader better navigate the system utilized.

- In Figure 1, labels may want to be unified to increase consistency. Fig 1a says interfollicular

- epidermis but Fig 1h says basal. Fig 1a has dermal fibroblasts but Fig 1h has pan-dermal fibroblasts.
- Fig. 1i and j seemed unclear in trying to convey by the diversity of ECM proteins. It might help to explain why it is chosen as the subject of study and cite relevant work suggesting the function of ECM diversity in an organ.
 - The criteria in transcriptome analysis may want to be clarified throughout to better support a reader and explain the reasons to choose the genes selected for instance in figure 1 through 3.
 - The manuscript wants to be checked for copyediting.

Reviewer #3 (Remarks to the Author):

The authors have combined bulk RNA-seq and immunohistochemical analysis to map out cell type and regional gene expression patterns in the skin basement membrane. Overall, while I found the RNA-seq analysis to be experimentally sound, the complete lack of statistical analysis in the downstream bioinformatics is a major concern. The authors obtained biological replicates for all of the regions and cell populations under study, but they really don't take advantage of the statistical power that this provides. As a result, the robustness and statistical significance of their findings are impossible to evaluate. In addition, the display items related to the RNA-seq data are unconventional, and I found them difficult to read and interpret:

1) In Fig. 1k, the authors show hierarchical clustering analysis of the RNA-seq data both using all genes detected above a certain threshold and a curated set of "matrisome" genes. My understanding is that the authors have biological replicate RNA-seq data sets for all of the cell populations clustered here, but the dendrograms only reflect the averaged expression values for each population. This analysis would be strengthened considerably if conducted at the level of individual samples or replicates rather than by averaging for each group of replicates. That way, the dendrogram would reflect not only how different cell populations co-cluster, but also how robust and consistent these relationships are across biological replicates.

2) Throughout the text, the authors assign gene sets as classifiers of various cell populations. There are too many cases to list here, but for example "Eighteen BM genes (e.g. Lama1, Lama3, Lama5, Col4a3, Col4a4, Col4a5, Col4a6, Col17a1, Eglf6, Fras1, Frem2, Npnt) were exclusively or predominantly expressed in the epidermal stem/progenitor populations (Fig. 2, Supplementary Table 2)." While claims like this may be true, the authors do not substantiate them with statistical analysis, and so I cannot evaluate their significance. As far as I can tell, the authors do not apply any statistical differential expression analysis to their RNA-seq data. Right now, the authors have used fragments per kilobase per million mapped reads (FPKM) as a proxy for relative expression levels throughout the paper. While this is fine for visualization of the data, I would recommend that the authors conduct a formal, count-based differential expression analysis (e.g. with something like DESeq2 or edgeR) in order to determine whether the expression patterns they have observed are statistically significant and to account for differences in coverage between genes and samples. For making comparisons involving multiple co-variates, one can also construct generalized linear models with, for example, DESeq2 to carry out statistical analysis. This is critical because even cases where genes are "exclusively or predominantly expressed" in specific regions or populations, if the number of counts is too low or unstable across replicates, these patterns may not be statistically significant.

3) Related to point 2), there are some thresholding-type analyses conducted by the authors in lieu of

actual statistical differential expression analysis. For example, “We identified cell populations that express certain ECM genes significantly more than other cell populations based on the following criteria: a gene expression level in a cell population i) of FPKM value ≥ 3 and ii) more than 40% of the FPKM value of the most highly expressed cell population.” These are arbitrary thresholds applied after normalization of the raw data that do not reflect the counting noise in RNA-seq and robustness of expression patterns across replicates. This is not an acceptable substitute for an actual statistical analysis on the raw count data, and I would recommend against this approach.

4) I found some of the main text figures in which the RNA-seq data are presented to be very complicated and difficult to interpret. For example, Figs. 3-4 contain large Venn diagrams that enclose numerous multi-color, multi-sized pie graphs labeled with gene names. I suspect that most of the patterns depicted here could be represented by a heatmap where overlap across different populations or sample groups can be readily visualized by sample and gene groupings across rows and columns and expression levels based on the color bars.

Point-by-point response to reviewers' comments (Tsutsui et al.; NCOMMS-20-09952)

REVIEWER COMMENTS

Reviewer #1 (Remarks to the Author):

This manuscript provides new insights based on an exceptionally broad, comprehensive characterization of ECM proteins comprising the microenvironments at separate sites of hair follicles. The main contributions include introduction of novel, exciting concepts regarding hook-and-mesh organization of basement membranes in developing tissues, as well as cell-basement membrane interaction at the HG-DP interface. These findings might prove helpful in developing regeneration strategies. The contributions of this paper could become sufficient for Nature Communications if the documentation were to be made stronger in a revised manuscript.

Specific concerns include the need to use more up-to-date RNA-seq analysis methods, clarification of methods and ruling out possible fluorescence artifacts, and clearer description of the Lama5-deletion mouse phenotype.

We thank the reviewer for the supportive and constructive comments.

Major points

1. RNA-seq analysis methods: The authors should use more up-to-date methods because the use of outdated software can affect results and reproducibility. For example, Tophat and Cuffdiff not only have known issues but their use in bioinformatics has been overtaken by newer software such as HISAT2 and STAR (Dobin et al., Bioinformatics 2013). Once familiarized with their use, applying newer methods to re-analyze the data should not take much time and might resolve some of the discrepancies between RNA-seq and protein localization noted below.

A specific issue is that a FPKM-based normalization method is not ideal for differential expression analysis. RPKM/FPKM methods introduce a bias in the per-gene variances, particularly for genes expressed at low levels. Please use newer raw count-based methods like DESeq2 and EdgeR instead.

<https://bioconductor.org/packages/release/bioc/html/DESeq2.html>

<https://bioconductor.org/packages/release/bioc/html/edgeR.html>

We thank the reviewer for pointing out these issues. In accordance with the reviewer's suggestion, we have now used more up-to-date methods, namely, Trim Galore, HISAT2 and StringTie, to obtain a matrix of read counts. We have also performed differential expression analysis with DESeq2 using the obtained read count matrix. To identify ECM genes that show differences in expression across hair follicle (HF) regions, we have performed the Wald test and likelihood ratio test for two-group comparison and multi-group comparison, respectively. To further identify sets of genes that exhibit similar expression patterns across sample groups, differentially expressed ECM genes (adjusted p -value < 0.001) were clustered by hierarchical clustering. The results are now described in Figs. 3 and 4. (page 10, lines 165–171 and page 12, lines 208–211 ; Methods, page 41, line 797–page43, line 846).

2. Discrepancies between RNA expression and protein localization, particularly in Supplementary Fig. 3: The authors are to be commended for their comprehensiveness and completeness in showing immunohistochemistry staining along with mRNA localization, which provide valuable complementary resources. However, the discrepancies between some RNA-seq results and protein localization seem quite confusing. This confusion results in part from a lack of presentation of these discrepancies in the Results text and only in the Discussion, from how the images are presented (staining does not always conform to the description of what it's supposed to stain, and discussion of the significance of the discrepancies. Although their identification of multiple key markers at the mRNA and protein levels in which these two expression results agree is obviously valuable, this level of confusion for other putative markers is confusing.

Aside from the molecules that the authors mention in the Discussion to have inconsistent localization patterns, other proteins also show inconsistent staining patterns that should be mentioned. For example:

- Mid bulge-specific: Ltbp3 is positive in dermal papilla. Collagen VI stains along the upper bulge, mid-bulge, and hair germ regions.*
- Hair germ-specific: Spondin-1 appears to be more dermal papilla-specific. No specific staining pattern of Decorin is evident.*
- Upper bulge & hair germ: Fibronectin stains hair germ and dermal papilla and not upper bulge. No hair germ staining is shown with Peroxidasin.*
- Dermal papilla-specific: No obvious staining pattern is observed with Col6a5 except for the basal layer.*

Potential reasons for these discrepancies need further elaboration. The RNA-seq

analysis methods as well as inherent methodological differences between RNA-seq and immunohistochemistry analyses may also account for this phenomenon.

*However, some potential autofluorescence artifacts are evident in what appears to be hair shafts (e.g., *Col17a1*, *Col13a1*, *Smoc1*). Were autofluorescence controls used to rule out false positives for these samples?*

We apologize for our confusing description on the discrepancy between RNA and protein. We have revised the manuscript by addressing the following three issues raised by the reviewer, in addition to reanalysing RNA-seq data: i) image presentation in Fig. S3, ii) lack of presentation of discrepancies in the Results text and iii) discussion of the significance of the discrepancies.

First, reanalysis of RNA-seq data gave us results of mRNA expression patterns of ECM genes similar to those of our original analysis. However, some of the discrepancies were solved (regarding *Lama2*, *Lama4*, *Fn1* and *Col6a5*). This would be partly due to the change in the method of categorization of gene expression patterns and the introduction of statistical analysis (please see our response to Reviewer #3).

i) Regarding the image presentation in Fig. S3, we consider that the labelling of the categorized ECM protein staining images in original Fig. S3 could be misleading. In the original manuscript, we categorized protein staining images based on their mRNA expression patterns. Therefore, for example, although LTBP3 protein is positive in both mid-bulge and DP, the protein staining image of LTBP3 was categorized to the mid-bulge-specific group based on its mRNA expression pattern. This is because we intended to show information about both the cellular origin of mRNA and the protein localization patterns of the ECM product together. To avoid potential misunderstanding, we now clearly label that the images are categorized by their ‘mRNA expression patterns’ in the ‘epithelium’ and ‘dermis’ (Fig. S5). For this purpose, we now introduce the term ‘epi-group’ to clearly indicate that these ECM gene groups were categorized by the analyses among epithelial cell populations. We have also confirmed that the signals from hair shafts were not from primary antibodies. These nonspecific signals are indicated by asterisks. As in our original image quantification, we did not include these nonspecific signals from the measurement, so we now describe our signal quantification methods in more detail (Fig. S5c and Methods; page 47, line 947–page 48, line 964).

Another potential cause of the mRNA–protein discrepancy is our separate mRNA data analyses for epithelial and dermal cell populations. We have separately analysed them because some of the highly expressed ECM genes in the dermal cell populations compromised the analysis of their differential expression among the

epithelial cell populations and vice versa. In contrast, immunostaining detects ECM proteins from a variety of cells and tissues, including nontarget cells in mRNA analysis, potentially causing discrepancy between mRNA and protein localization. Thus, we added information about the expression-level predominance between the basal epithelial population and pan-dermal fibroblasts to help the reader to consider the contribution of epithelial- and dermal-derived ECM proteins in tissue localization (Fig. 5b, c; page 8, line 138–page 10, line 160).

ii) Regarding the lack of presentation of discrepancies between mRNA and protein, we have now prepared new Table S3, which provides the data for considering consistency and discrepancy between mRNA and protein localization. In this table, we also describe possible causes of the discrepancies. Some of the ECM molecules showing a discrepancy are also described in Results (page 14, lines 233–236).

iii) For discussion of the significance of the discrepancies, we have elaborated the potential reasons for these discrepancies by discussing the biological complexity in regulating the level of ECM proteins in tissues and also inherent methodological differences between RNA-seq and immunohistochemical analyses (page 26, lines 398–415).

3. With the Lama5 conditional knockout mouse model showing a non-essential phenotype for hair growth, Lama5 being required for “functional integrity of hair germ–dermal papilla interactions” is a vague possible overstatement. The wording should be more precise here.

We agree with the reviewer. Because we obtained new data showing that *Lama5* is required for hair cycle regulation and hair germ (HG)-dermal papilla (DP) anchoring, we have changed the wording accordingly (page 2, lines 33–34, page 5, lines 95–97, page 22, lines 337–340 and page 24, lines 367–368).

4. The findings of mesh/hook BM, BM protrusions and cell-BM interaction at the HG-DP interface are quite interesting. More insights and, if possible, some additional speculation about their significance would be helpful. Also, please add a recent paper (Ge...Fuchs et al., PNAS 2020) to your Discussion. They provide single-cell molecular characterization of murine hair follicles and show differences in ECM genes between young and old hair follicle stem cells. How do your findings compare to theirs?

We appreciate the reviewer's suggestions. To gain more insights into the significance of cell interaction with hook and interface BMs (and also to respond to Reviewer #2's similar comments), we first examined the deposition dynamics of two hook and interface BM components, laminin $\alpha 5$ and COL13A1, during the hair cycle because they play critical roles in the adhesion and maturation of presynaptic and postsynaptic specializations in the neuromuscular junctions (Rogers and Nishimune, 2016; Latvanlehto et al., 2010). These proteins were increased when the surface area of the HG–DP interface was small (telogen) and upward movement of the HF epithelium and DP occurred (catagen), but decreased when DP was fully enclosed by the HF epithelium (anagen IIIb), suggesting the functional importance of hook and interface BMs in anchoring the HG and DP (Fig. 9a; page 22, line 346–page 24, line 354).

We next investigated the expression patterns of $\beta 1$ integrin, a major receptor of laminin $\alpha 5$, in the DP. $\beta 1$ integrin was enriched at the interface and hook BMs when laminin $\alpha 5$ deposition was increased (Fig. 9a; page 24, lines 354–355). Deletion of epithelial *Lama5* depleted the accumulated $\beta 1$ and $\alpha 6$ integrins from the hook and interface BMs (Fig. 8b; page 20, lines 306–310). These results suggest that epithelial-derived laminin $\alpha 5$ plays critical roles in the integrin-mediated BM–DP interactions.

We further investigated the connectivity of the HG–DP interface in conditional *Lama5* knockout mice at second telogen (P84). We found that mutant HFs have smaller DP and occasionally show dispersed or detached DP (Fig. 9b, c; page 24, lines 359–368), indicating the role of epithelial-derived *Lama5* in HG–DP anchoring.

In response to another reviewer's comment, we also examined signalling defects in *Lama5* mutant. We found that SMAD2, which is activated at late telogen in response to TGF- $\beta 2$ produced by the DP to induce anagen entry, was precociously activated in the HG of early telogen HFs of mutant mice, providing a possible role of cell–BM interactions in regulating signalling exchange between HG and DP to regulate the hair regeneration cycle (Fig. 8f; page 20, line 320–page 22, line 331). The expression of Wnt co-transcription factor LEF1 was also greatly diminished in the DP, suggesting that the defects in BM–DP interactions abolish DP cell identity (Fig. 8g; page 22, lines 332–340).

Taken together, our new results suggest that hook and interface BMs are involved in the anchoring of HG and DP as well as the signalling exchange between them. These BM-mediated HG–DP interactions are required for regulating the HF regeneration cycle. We have now included these new results in the revised manuscript.

Regarding the comparison between the findings in our study and those of Ge et al. (2020), we used their single-cell RNA-seq data to i) examine the consistency of our

data with their single-cell data from young epithelia, and ii) investigate age-associated ECM gene expression changes by focusing on HG and bulge cells. The expression patterns of most of the differentially expressed ECM genes were consistent with our results (Fig. S9a; page 29, lines 481–482). One major discordance was the absence of 15 HG-specific ECM genes (epi-group 6) in the list of differentially expressed genes created using the scRNA-seq dataset of Ge et al. (Fig. S9a, b; page 29, lines 482–486). This difference is potentially due to the cell population bias resulting from the difference in cell isolation protocol and RNA-seq procedure. We further compared their gene expression between young and aged HG and bulge populations and identified significantly differentially expressed ECM genes (Fig. S9c, d; page 29, lines 486–page 30, lines 493). ECM genes involved in the epithelial interaction with mesenchyme are listed as age-associated downregulated genes in both HG and bulge, suggesting that BM-mediated epithelial–mesenchymal interactions are altered in aged HFs. We have now included these new results in the revised manuscript.

Minor points

5. It is puzzling that the authors do not present a more-detailed hierarchical clustering analysis, e.g., in supplemental material, showing each of the separate biological replicates rather than just pooled data. Although the analysis shown in the main text is of course useful conceptually, this type of detailed hierarchical clustering analysis can provide readers with insight into the level of variability between different biological replicates.

We agree. We have replaced our original hierarchical clustering data in Fig. 1 with those with individual biological replicates (Fig. S1n; page 6, lines 111–114). We have also visualized gene expression patterns of individual replicates in Figs. 3, 4, S2 and S4.

6. Fig. 6g: Hemidesmosomes should be indicated by white arrows in the lateral BM as well, in order to compare with interface BM.

We agree. We have added white arrowheads to indicate hemidesmosomes in the lateral BM (Fig. 6g).

7. Please consider whether some detailed methods information presented in Results is needed. For example, -spearman's rank correlation coefficient analysis... (Line 113-114).

-“...based on the following criteria: a gene expression level in a cell population i) of FPKM value ≥ 3 and ii) more than 40% of the FPKM value of the most highly expressed cell population. Categorized genes were depicted using a Venn diagram... We identified a commonly expressed epidermal stem/progenitor ECM gene group at the middle of the Venn diagram” (Line 182-187).

We have revised this method information and moved it to the Method part.

8. Minor typographical errors:

-The first letter of mouse proteins should be capitalized.

In accordance with the International Protein Nomenclature Guidelines (https://www.ncbi.nlm.nih.gov/genome/doc/internatprot_nomenguide/), we use an all-uppercase gene symbol in an acronymic protein name, or an all-lowercase name for a functional protein name for mouse protein names (please read Part 2, B and E, of the guidelines).

-Sentence structure of headings should be consistent. Many headings are sentence fragments whereas others use complete sentence structure.

We have made the sentence structure of headings consistent.

-The use of word “despite” was confusing at times.

-Line 104: add comma after (integrin $\alpha 6$ +))

-Line 181: “cell-type-dependent”  “cell type-dependent”

-Line 201: “mid-bulge-“  “mid bulge-“

-Line 378: synthesis locally  local synthesis

We have corrected these typographical errors accordingly. However, as per our proofreader’s advice, we have left a hyphen in “mid-bulge” in all instances, as “mid” is not a complete word but a prefix, so should be linked to the subsequent text.

Reviewer #2 (Remarks to the Author):

In the manuscript, Tsutsui et al addressed the molecular and spatial details of the

basement membrane using quantitative transcriptomics and immunofluorescence. They used mouse telogen skin as a model to determine the expression profiles of ECM molecules from different epidermal/dermal compartments in homeostasis. This work provides a comprehensive picture of the cellular contributions from epithelial cells and fibroblasts, adding quantitative details to our understanding of the basement membrane. Specifically, the authors identified region-specific expression patterns, providing foundation for the niche-specific basement membrane architecture. To further their spatial understanding, the authors examined a total of 41 protein patterns via antibody immunostaining with outstanding clarity. They first interrogated the protein source by comparing protein with mRNA expression patterns, revealing mostly similarity that indicates local assembly as well as some discrepancy suggesting long-range transportations. Then they focused on the basement membrane between the epithelial hair germ and the mesenchymal DP, and identified the “hook” and “mesh” structures. Lastly, they showed the importance of one “hook” protein, laminin a5 to hair follicle structure and function. The data presented here are a beautiful combination of comprehensive transcriptome analysis and in-depth analysis of a regional BM spatial and functional details. These findings are of high importance and interest and will represent an important reference to the field moving forward. Below a number of suggestions for the authors to consider to improve coherence and/or accuracy where needed.

We thank the reviewer for the favourable and constructive comments.

- The focus on basement membrane is obscured in the transcriptomic analysis, since the authors put much emphasis on interstitial ECM as well. Perhaps the interstitial ECM could serve as a control instead of a counterpart for the coherency of the story. For example, the authors used all matrisome for hierarchy clustering. It was not clear whether basement membrane matrisome only will have stronger/similar clustering. Additionally why pan-dermal fibroblasts don't cluster with basal epithelial cells: does that indicate distinct regulatory mechanisms involving basement membranes of different locations?

We agree with the reviewer that the focus on BM is obscured in the transcriptomic analysis. To maintain focus, we have reduced the description of interstitial ECMs in the Results section and moved some main figures related to the interstitial ECMs to Figs. S2 and S3.

Thank you for suggesting the performance of clustering with BM matrisome only. Before implementing this analysis, we have changed the analytical source data from FPKM values to normalized read count data in response to the other reviewers' comments. We first performed hierarchical clustering, PCA and Spearman's rank correlation coefficient analysis using all matrisome genes. Although HG cells and DP cells showed strong correlations in Spearman's rank correlation coefficient analysis, HG cells did not cluster with DP cells in the newer hierarchical clustering, possibly due to slight changes in the hierarchical relationship between samples (we still see the same trend as in our analysis with FPKM values). We have also used BM genes, but HG and DP did not form a cluster in hierarchical clustering. Because Spearman's rank correlation coefficient analysis provides correlation values between two samples, we have used this for further analysis. As mentioned above, when all matrisome genes were used, HG cells and DP cells showed a strong correlation (Fig. 1i; page 6, line 119–page 8, line 136). We then used BM genes or interstitial ECM genes and found that interstitial ECM genes contribute to the higher correlation between HG and DP ECM expression profiles than BM genes.

As pointed out by the reviewer, the pan-dermal fibroblast population did not cluster with the basal epithelial cell population, while DP clustered with the counterpart epithelia, HG. As mentioned by the reviewer, this could be explained by the distinct regulatory mechanisms of BM formation used in different skin locations. Another potential reason would be the difference in the homogeneity/specificity of examined cell populations. DP and HG populations were isolated from a tightly facing local epithelial–mesenchymal interaction site. In contrast, pan-dermal fibroblasts are a mixture of *Pdgfra*⁺ dermal cells from all regions of the dorsal dermis, including cells far from the BM. Similarly, the basal epithelial cell population is a mixture of cells from all dorsal epithelial regions. Because both epithelial cells and fibroblasts in the skin show high cellular heterogeneity (Donati et al., *Cell Stem Cell* 2015; Rognoni et al., *Trends in Cell Biol* 2018), low spatial specificity and contiguity of cell populations in our pooled populations might flatten the specific gene expression and thus could be one cause of the weaker clustering. Single-cell-level transcriptome analysis may allow us to compare ECM gene expression of epithelial and dermal cell populations that attach to distinct shared BMs. We have mentioned a future direction of study toward single-cell ECM transcriptomic analysis in the revised manuscript (page 25, line 379–page 27, line 423).

- *Could the dataset in Figure 2 be more quantitative about the actual contributions from fibroblasts and epithelial cells? Would be helpful if the authors have a way to quantify*

the percentage of expressions between epithelial cells and fibroblasts

Suitable cell populations to be used for this comparison in our datasets would be ‘basal epithelial cells’ and ‘pan-dermal fibroblasts’, which contain averaged gene expression data of mixtures of pan-epidermal basal cells and pan-dermal fibroblasts, respectively. We have examined the ratio of ECM gene expression levels in these cell populations with normalized gene count data and replaced our original Table S2 with bar graphs showing the new quantitative data (Figs. 2 and S3; page 8, line 138–page 10, line 160). These analyses provide a quantitative and comprehensive understanding of the contributions of epithelial cells and fibroblasts to BM and interstitial ECM gene expression.

- It was not clear how the “rigorous validation for specificity” (line 241) for antibody staining is performed.

Thank you for pointing this out. We have now added a detailed description of our antibody validation in the Methods part (page 46, line 927–page 47, line 934). We have also removed the word ‘rigorous’ because the assessment of antibody specificity could be more rigorous if we combine other methods (page 14, line 219–221).

- The name “hook” is intriguing as it matches its appearance but may be leading to the assumption that the BM actually function as a hook to anchor DP to the hair germ. Additionally, could there be an opportunity to extract more information from their exciting laminin $\alpha 5$ knock-out model to understand the function of the “hook” BM? For example, were there any detachment of DP from hair germ in catagen for laminin $\alpha 5$ knock-out strains (if the hook is not working)? What is the cause of the hair cycle defect? Hair follicles appeared to have more DP cells in Lama5^{-/-}. Was that something consistent among mice and how do the authors explain this?

We appreciate these valuable comments. We have further examined the functional aspects of the hook BM by utilizing laminin $\alpha 5$ cKO mice (Reviewer #1 also pointed out similar issues). First, we have examined the expression patterns of two hook BM molecules, laminin $\alpha 5$ and COL13A1, in the hair cycle because they play critical roles in the adhesion and maturation of presynaptic and postsynaptic specializations in the neuromuscular junctions (Rogers and Nishimune, 2016; Latvanlehto et al., 2010). We found that these proteins are increased when the surface area of the HG–DP interface is

small (telogen) and upward movement of HF epithelium and DP occurs (catagen), but decreased when DP is fully enclosed by the HF epithelium (anagen IIIb), suggesting the functional importance of hook and interface BMs in anchoring the HG and DP (Fig. 9a; page 22, line 346–page 24, line 354).

We next investigated the expression patterns of $\beta 1$ integrin, a major receptor of laminin $\alpha 5$, in the DP. $\beta 1$ integrin was enriched at the interface and hook BMs when laminin $\alpha 5$ deposition was increased (Fig. 9a; page 24, lines 354–355). Deletion of epithelial *Lama5* depleted the accumulated $\beta 1$ and $\alpha 6$ integrins from the hook and interface BMs (Fig. 8b; page 20, lines 306–310). These results suggest that epithelial-derived laminin $\alpha 5$ plays critical roles in the integrin-mediated BM–DP interactions.

We further investigated the connectivity of the HG–DP interface in conditional *Lama5* knockout mice at second telogen (P84). We found that mutant HFs have smaller DP and occasionally show dispersed or detached DP (Fig. 9b, c; page 24, lines 359–368), indicating the role of epithelial-derived *Lama5* in HG–DP anchoring.

In regard to the cause of the hair cycle defects, we examined the activity of the TGF- β /SMAD2 signalling pathway in the HG, which is activated in late telogen in HG in response to TGF- β secretion from the DP and induces hair cycle entry. In conditional *Lama5* knockout mice, SMAD2 was activated at early telogen, well before the anagen entry (Fig. 8f; page 20, line 320–page 22, line 331), suggesting that TGF- β signalling in the HG of mutant HFs is highly activated throughout early to late telogen phases. The expression of LEF1 was also greatly diminished in the DP of the mutant HFs, suggesting the defects in DP cell identity and Wnt signalling in the HG–DP interface (Fig. 8g; page 22, lines 332–340). Taken together, our new results suggest that hook and interface BMs are involved in the physical connection between HG and DP as well as their signalling exchange, underlying, at least in part, dysregulated hair cycle progression in *Lama5* cKO mice. We have now included these new results in the revised manuscript.

We agree about apparent increase of DP cells in the developing whisker follicles in *Lama5*^{-/-} mice. However, we decided to measure the DP size in adult dorsal HFs because we have accumulated new data with adult dorsal HFs and thus considered that it would be better to focus our study on HF regeneration. We found that the mutant HFs have smaller DPs than controls (Figs. 9b, c; page 24, lines 359–368). Some small DPs were detached or dispersed from the HG, indicating that epithelial-derived laminin $\alpha 5$ plays a role in anchoring the DP to the HG and in regulating DP size. We have transferred data of developing HFs to the supplementary data to maintain focus (Figs. S9a, b).

- The conclusion of continuous basement membrane (line 329) may not be always supported by some of the staining in figure 7. One can also imagine that two populations of BM in close proximity will also be revealed as connected in an immunostaining image, where the resolution is low. The captured EM data may provide some more opportunities to address this question?

We acquired super-resolution three-dimensional immunofluorescent images of the interface BM (Supplementary Video1). Although some single-focal-plane images showed two apparently separated populations of BM in close proximity around the interface BM, their connection can be found in other neighbouring focal planes. We have described this in the revised manuscript (page 17, lines 262–264).

Minor points:

- The introductions lack the context of telogen hair follicles. Adding references about the BM components and functional studies in DP may help a reader better navigate the system utilized.

We have added an introduction to the hair cycle and several references about BM components and their functional studies in DP (page 4, lines 62–66 and page 5, lines 85–87).

- In Figure 1, labels may want to be unified to increase consistency. Fig 1a says interfollicular epidermis but Fig 1h says basal. Fig 1a has dermal fibroblasts but Fig 1h has pan-dermal fibroblasts.

We have unified these terms to ‘basal epithelial cells’ and ‘pan-dermal fibroblasts’. We have also now distinctly used ‘epithelium’/‘epithelial’ and not ‘epidermis’/‘epidermal’ when describing the HF epithelium. We have received a suggestion from a researcher that the epidermis by definition is the interfollicular epithelium only and separate from the HF epithelium.

- Fig. 1i and j seemed unclear in trying to convey by the diversity of ECM proteins. It might help to explain why it is chosen as the subject of study and cite relevant work suggesting the function of ECM diversity in an organ.

We agree. We have reconsidered the necessity of Figs. 1i and j in light of the contents of our revised manuscript and decided to remove them.

- The criteria in transcriptome analysis may want to be clarified throughout to better support a reader and explain the reasons to choose the genes selected for instance in figure 1 through 3.

In response to other reviewers' comments, we have introduced several new statistical criteria in our transcriptome analysis to define significantly differentially expressed genes. In addition, to select genes preferentially expressed in certain cell populations, we have introduced hierarchical clustering, k-means clustering and fold change thresholding. We believe that these revised procedures make the criteria of gene selection clearer and more statistically meaningful. Please see revised Figs. 1–4, S3 and S4.

- The manuscript wants to be checked for copyediting.

We have asked for careful copyediting of the manuscript by a professional English editing service.

Reviewer #3 (Remarks to the Author):

The authors have combined bulk RNA-seq and immunohistochemical analysis to map out cell type and regional gene expression patterns in the skin basement membrane. Overall, while I found the RNA-seq analysis to be experimentally sound, the complete lack of statistical analysis in the downstream bioinformatics is a major concern. The authors obtained biological replicates for all of the regions and cell populations under study, but they really don't take advantage of the statistical power that this provides. As a result, the robustness and statistical significance of their findings are impossible to evaluate. In addition, the display items related to the RNA-seq data are unconventional, and I found them difficult to read and interpret:

We appreciate the reviewer's critical comments and constructive suggestions.

1) In Fig. 1k, the authors show hierarchical clustering analysis of the RNA-seq data

both using all genes detected above a certain threshold and a curated set of “matrisome” genes. My understanding is that the authors have biological replicate RNA-seq data sets for all of the cell populations clustered here, but the dendrograms only reflect the averaged expression values for each population. This analysis would be strengthened considerably if conducted at the level of individual samples or replicates rather than by averaging for each group of replicates. That way, the dendrogram would reflect not only how different cell populations co-cluster, but also how robust and consistent these relationships are across biological replicates.

We thank the reviewer for this important comment. We have performed hierarchical clustering analysis with individual biological replicates of each population using normalized gene count data calculated by DESeq2. The new results show that biological replicates co-cluster, but do not cluster with other populations (Fig. S1n). We have also performed PCA again with normalized gene count data (Fig. S1m).

2) Throughout the text, the authors assign gene sets as classifiers of various cell populations. There are too many cases to list here, but for example “Eighteen BM genes (e.g. Lama1, Lama3, Lama5, Col4a3, Col4a4, Col4a5, Col4a6, Col17a1, Eglf6, Fras1, Frem2, Npnt) were exclusively or predominantly expressed in the epidermal stem/progenitor populations (Fig. 2, Supplementary Table 2).” While claims like this may be true, the authors do not substantiate them with statistical analysis, and so I cannot evaluate their significance. As far as I can tell, the authors do not apply any statistical differential expression analysis to their RNA-seq data. Right now, the authors have used fragments per kilobase per million mapped reads (FPKM) as a proxy for relative expression levels throughout the paper. While this is fine for visualization of the data, I would recommend that the authors conduct a formal, count-based differential expression analysis (e.g. with something like DEseq2 or edgeR) in order to determine whether the expression patterns they have observed are statistically significant and to account for differences in coverage between genes and samples. For making comparisons involving multiple co-variates, one can also construct generalized linear models with, for example, DEseq2 to carry out statistical analysis. This is critical because even cases where genes are “exclusively or predominantly expressed” in specific regions or populations, if the number of counts is too low or unstable across replicates, these patterns may not be statistically significant.

We thank the reviewer for pointing out this critical issue and providing constructive

suggestions. In accordance with these suggestions, we have performed statistical differential expression analysis with DEseq2 using count-based expression data, but not FPKM values. Please see the procedure of the preparation of count-based gene expression matrix in the Method (page 41, lines 801–page 42, 813), as well as the response to Reviewer #1. To group these differentially expressed ECM genes based on their gene expression patterns across epithelial or dermal cell populations, we performed hierarchical clustering analysis (Figs. 3, 4; page 10, lines 165–171 and page 42, line 818–page 43, line 841). We cut the epithelial dendrogram at the height where it gave 10 sub-gene groups. To annotate ECM gene expression patterns in each sub-gene group, the epithelial regions were statistically divided into two groups based on the expression patterns of the clustered ECM genes by *k*-means clustering. The cell group expressing the cluster-composing ECM genes at a level higher than the other group was classified as epithelial regions showing relatively high expression of cluster-composing ECM genes (please see “Pattern” in Fig. 3). We believe that these revised procedures allow us to classify gene expression patterns with statistical significance. Please read revised Fig.s 1–4, S3 and S4.

3) Related to point 2), there are some thresholding-type analyses conducted by the authors in lieu of actual statistical differential expression analysis. For example, “We identified cell populations that express certain ECM genes significantly more than other cell populations based on the following criteria: a gene expression level in a cell population i) of FPKM value ≥ 3 and ii) more than 40% of the FPKM value of the most highly expressed cell population.” These are arbitrary thresholds applied after normalization of the raw data that do not reflect the counting noise in RNA-seq and robustness of expression patterns across replicates. This is not an acceptable substitute for an actual statistical analysis on the raw count data, and I would recommend against this approach.

We agree. Please see our response to comment #2, which covers this issue.

4) I found some of the main text figures in which the RNA-seq data are presented to be very complicated and difficult to interpret. For example, Figs. 3-4 contain large Venn diagrams that enclose numerous multi-color, multi-sized pie graphs labeled with gene names. I suspect that most of the patterns depicted here could be represented by a heatmap where overlap across different populations or sample groups can be readily visualized by sample and gene groupings across rows and columns and expression

levels based on the color bars.

Thank you for your valuable suggestion. We have now used hierarchical clustering to make groups of genes and represented their gene expression patterns by a heatmap. Please see our response to comment #2, which covers this issue.

REVIEWERS' COMMENTS

Reviewer #1 (Remarks to the Author):

The authors have responded unusually conscientiously and comprehensively to the original review. For example, indicating the discrepancies and agreements in Table S3 is valuable for completeness. They have resolved all original concerns identified. Consequently, this intriguing paper is now appropriate for acceptance for publication.

Reviewer #2 (Remarks to the Author):

The authors have now added additional data and refined the analysis/representation to address reviewers' concerns, significantly improving its overall quality. The new RNA-seq analysis method allowed for the manuscript to nicely focus on BM, provided additional details, and re-defined the conclusions about the cluster. The functional analysis of hook BM revealed exciting science and helped strengthen the significance and the scientific rigor. With these additional work, this manuscript represents an exciting and comprehensive contribution that importantly move forward to the understanding of basement membrane biology in vivo. Thank you to the authors for these contributions.

Reviewer #3 (Remarks to the Author):

In my opinion, the manuscript and figures are greatly improved by the authors revisions. I have one comment:

In response to one of my previous criticisms, the authors have added Supplementary Figure 1n, which shows sample-level clustering of the data (rather than just clustering of averages for each condition). These results are very helpful and demonstrate the overall reproducibility of their measurements, but this analysis is presumably using all genes. It would be great to show this same type of analysis for the more focused gene sets. For example, it would be great to show the sample-level results for the heatmaps in Fig. 1h-i rather than just averages over all of the samples for each condition. This would visually display the robustness of the major trends.

Point-by-point response to reviewers' comments (Tsutsui et al.; NCOMMS-20-09952B)

Mapping the molecular and structural specialization of the skin basement membrane for inter-tissue interactions

REVIEWER COMMENTS

Reviewer #3 (Remarks to the Author):

In my opinion, the manuscript and figures are greatly improved by the authors revisions. I have one comment:

In response to one of my previous criticisms, the authors have added Supplementary Figure 1n, which shows sample-level clustering of the data (rather than just clustering of averages for each condition). These results are very helpful and demonstrate the overall reproducibility of their measurements, but this analysis is presumably using all genes. It would be great to show this same type of analysis for the more focused gene sets. For example, it would be great to show the sample-level results for the heatmaps in Fig. 1h-i rather than just averages over all of the samples for each condition. This would visually display the robustness of the major trends.

Thank you for your suggestion. We have replaced the original Fig. 1h with the new figure that presents sample-level results (biological replicates). Regarding Fig. 1i, the original heatmap panels present individual biological replicate values. Because the variation of the data between replicates is very small, most of the replicates show quite similar heatmap cell colour. Thus, to make clear that the data present individual replicate values, we indicated the number of replicates in the figures and legends of both Fig. 1h and 1i and added support lines indicating the borders between the heatmap cells of Fig. 1i.